# `Motley`: Benchmarking Heterogeneity and Personalization in Federated Learning

**Shanshan Wu**[1], **Tian Li**[2], **Zachary Charles**[1], **Yu Xiao**[1],
**Ziyu Liu**[2], **Zheng Xu**[1], **Virginia Smith**[2]
[1]Google Research {shanshanw, zachcharles, xiaoyux, xuzheng}@google.com
[2]Carnegie Mellon University {tianli, kzliu, smithv}@cmu.edu

## Abstract

Personalized federated learning considers learning models unique to each client in a heterogeneous network. The resulting client-specific models have been purported to improve metrics such as accuracy, fairness, and robustness in federated networks. However, despite a plethora of work in this area, it remains unclear: (1) which personalization techniques are most effective in various settings, and (2) how important personalization truly is for realistic federated applications. To better answer these questions, we propose `Motley`, a benchmark for personalized federated learning. `Motley` consists of a suite of cross-device and cross-silo federated datasets from varied problem domains, as well as thorough evaluation metrics for better understanding the possible impacts of personalization. We establish baselines on the benchmark by comparing a number of representative personalized federated learning methods. These initial results highlight strengths and weaknesses of existing approaches, and raise several open questions for the community. `Motley` aims to provide a reproducible means with which to advance developments in personalized and heterogeneity-aware federated learning, as well as the related areas of transfer learning, meta-learning, and multi-task learning. Code for the benchmark is open-source and available at: https://github.com/google-research/federated/tree/master/personalization_benchmark.

## 1 Introduction

Federated learning (FL) aims to share knowledge across disparate data silos in a privacy-preserving manner [36, 46, 57]. Relative to standard (e.g., data center) distributed learning, a defining trait of federated learning is the presence of *heterogeneity*[3], as each of the clients (e.g., a sensor, mobile phone, or entire organization) may generate data according to a distinct distribution. In response to this key difference, a significant amount of work has been devoted to understanding and addressing heterogeneity in federated learning. For example, heterogeneity has been the focus of the analysis and development of numerous federated optimization methods [e.g., 1, 13, 29, 37, 47, 50, 72, 73, 74], and the impact of heterogeneity on FL has been explored more generally in connection with issues of fairness, robustness, and privacy in federated networks [e.g., 48, 49, 58, 79].

The presumed presence of heterogeneity has also directly driven a large body of work in *personalized federated learning*. In personalized FL, the goal is to learn or adapt models to more closely reflect the distinct data distribution of each federated client. For example, personalized FL approaches may consider fine-tuning a global or meta-trained model to adapt to local client data [16, 19, 35, 38, 67, 82], learning a clustering structure amongst the clients [20, 59, 65], or using multi-task learning techniques to model possibly more complex relationships [17, 21, 22, 49, 68].

Despite a significant amount of work in personalized FL, there remains a lack of consensus about which methods perform best in various federated settings, let alone the degree to which personalization

Workshop on Federated Learning: Recent Advances and New Challenges, in Conjunction with NeurIPS 2022 (FL-NeurIPS'22). This workshop does not have official proceedings and this paper is non-archival.

itself is really necessary in practice. For example, many works target a specific form of personalization (e.g., clustering) in their methodology, and then artifically create or modify data to match this assumption (e.g., manually clustering the data) before testing the efficacy of their approach [74]. While this is a reasonable sanity check, it fails to validate how impactful such approaches are for real-world FL applications. In particular, a major issue is that while 'heterogeneity' is believed to be a natural occurrence in federated networks, there is a lack of benchmarks that reflect real applications of FL, making it hard to understand the prevalence and magnitude of heterogeneity in practice.

To address these concerns, we propose `Motley`, a comprehensive benchmark for personalized and heterogeneity-aware FL. `Motley` is designed with a focus on ease-of-use and reproducibility[1]. The benchmark includes a suite of four cross-device and three cross-silo datasets, as well as baseline personalization methods evaluated on them. The datasets themselves are drawn from real-world applications mirroring federated settings in an effort to better reflect naturally occurring forms of heterogeneity. In developing baselines for the benchmark, we pay careful attention to the evaluation of personalization and heterogeneity—making concrete suggestions for future work in evaluating the impact of heterogeneity/personalization. Finally, although we focus specifically on the application of FL, this benchmark is also a useful tool for the areas of transfer learning, meta learning, and multi-task learning more generally, as techniques from these areas are commonly used for personalized FL.

**Cross-device vs. Cross-silo.** Based on the size and characteristics of the network, there are two common FL settings: *cross-device FL* and *cross-silo FL*, and two types of algorithms: *stateful* and *stateless* [74]. We provide a brief description below (see Table 1), and defer readers to [36, 74] for more detailed discussion. Cross-device applications typically involve learning across a large number (e.g., hundreds of millions) of mobile or IoT devices. Given the network scale coupled with the unreliability of such devices, it is common for devices to only participate once (if at all) in training. This characteristic motivates the development of methods for cross-device FL which are *stateless*[2], in that model or variable state is not maintained on each client from one round to another, and there exists no unique identifier for each client. In contrast, cross-silo FL applications often consider learning across a handful of organizations such as hospitals or schools, where clients are almost always available at each training round. These properties allow the silos to be more easily accessed and identified, permitting the use of possibly more complex *stateful* approaches.

**Table 1:** Differences between the cross-device and cross-silo experimental setups.

| Experimental setup | Cross-device | Cross-silo |
|---|---|---|
| Client sampling rate per round (Appendix A.2) | On the scale of 0.1%-1% (see Table 3 for the exact value) | 100% (all silos participate in every round) |
| Train/valid/test split (Section 2.2) | Split clients (Fig. 1) | Split client's data (Fig. 1) |
| Stateful or stateless algorithm | Stateless[2]. | Both work. |

**Heterogeneity and Personalization.** As pointed out previously, a defining trait of FL is that each client may generate according to a unique distribution. To better account for this heterogeneity[3], it is therefore common to consider techniques that learn personalized, client-specific models. Existing personalized FL approaches can be categorized in three different ways: (1) stateful vs stateless (see Table 1 and the discussion above); (2) model-agnostic vs model-specific (whether it targets a specific model and requires domain-specific information [e.g., 34, 51, 67, 84]); (3) in terms of methodologies, e.g., meta-learning [2, 13, 19, 35, 38], clustering [20, 53], local memory [54], and multi-task learning (MTL) [e.g., 4, 17, 21, 22, 23, 49, 52, 68]. We defer readers to Appendix D and the recent surveys [70], [74, §7.5] for more detailed discussion on existing literature. In benchmarking `Motley`, we consider five model-agnostic personalized FL methods (four stateless and one stateful) from the major methodology categories (meta-learning, clustering, local memory, and MTL).

---

[1]Code for the benchmark is open-source and available at: https://github.com/google-research/federated/tree/master/personalization_benchmark.

[2]Stateful algorithms can perform poorly in the cross-device settings where the clients sampling rate is very low in each round. Because most clients are sampled at most once during the entire training process, their local states are either unused or very stale. See Section 5.1 of [62] for the empirical results of the stateful SCAFFOLD algorithm [37] in the cross-device setting.

[3]We focus on *data heterogeneity* and do not consider other types of heterogeneity in our experiments, e.g., mobile devices have different processing speeds and memory constraints, which is worth exploring in the future.

**Table 2:** `Motley` has three components: (1) modular implementations of five representative personalized FL algorithms, (2) a diverse range of tasks (C: classification; R: regression; NWP: next word prediction) and datasets (see Appendix E) chosen to cover the cross-device and cross-silo FL settings, and (3) baseline results via extensive hyparameter tuning (see Appendix F) and the insights. The right-most column is the average number of examples per client $\pm$ standard deviation.

| Methods | Dataset Details | | Clients | Pts/Client |
|---|---|---|---|---|
| | *Dataset* | *Task and Model* | | |
| *Cross-Device FL* | | | | |
| Local training | EMNIST [6] | Image C; CNN | 3400 | 198±89 |
| FedAvg+Fine-tuning [35] | StackOverflow [7] | NWP; LSTM | 380k | 397±1279 |
| HypCluster [53]/IFCA [20] | Landmarks [30] | Image C; MobileNetV2 | 1262 | 130±199 |
| FedAvg+kNN-Per [54] | TedMulti-EnEs [61] | NWP; Transformer | 4184 | 113±56 |
| *Cross-Silo FL* | | | | |
| Local training | ADNI[5] | Image R; CNN | 9 | 5405±4822 |
| FedAvg+Fine-tuning [35] | Vehicle [18] | Binary C; SVM | 23 | 1900±349 |
| HypCluster [53]/IFCA [20] | School[6] | R; Linear Regression | 139 | 111±56 |
| FedAvg+kNN-Per [54] | | | | |
| MTL [49, 68] | | | | |

**Existing FL benchmarks.** Concurrent works benchmarking personalized FL [15, 55] do not take into account differences between cross-device and cross-silo federated settings (Table 1), which we find to have a significant impact on the baseline methods and resulting conclusions. Beyond these works, other prior FL benchmarks (see, e.g., [8, 9, 10, 12, 26, 27, 31, 42, 44]) do not consider personalization baselines and focus instead on the standard FedAvg [57] algorithm and variations [62].

## 2 `Motley`: A Benchmark for Personalized Federated Learning

Table 2 gives an overview of the three components of `Motley` (methods, datasets, baselines). We now briefly describe the methods and datasets. See Appendix A for more detailed description of `Motley`.

### 2.1 Personalization Methods

`Motley` includes five simple and model-agnostic algorithms for learning personalized models: (1) **Local training**, refers to every client training a local model using their own data, without collaborating with others. (2) **FedAvg+Fine-tuning** [35], in which we first train a global model via FedAvg [62][4], and then, each client fine-tunes the global model on their local data to get the personalized model. This simple method has a natural connection to meta-learning [35]. (3) **HypCluster** [53] (also **IFCA** [20]), which jointly clusters clients and learns a model for each cluster. (4) **FedAvg+kNN-Per** [54], a very recent personalization algorithm proposed to interpolate/ensemble the output of two models: a globally-trained FedAvg model and a local kNN model. (5) **Multi-Task Learning** (MTL), a class of methods used to deliver personalized models for a set of tasks by learning the task relations (either explicitly or implicitly). The first four algorithms are *stateless*[2], and hence, are appropriate in the cross-device FL setting (see Table 1 and [36, 74] for discussions on stateful vs stateless).

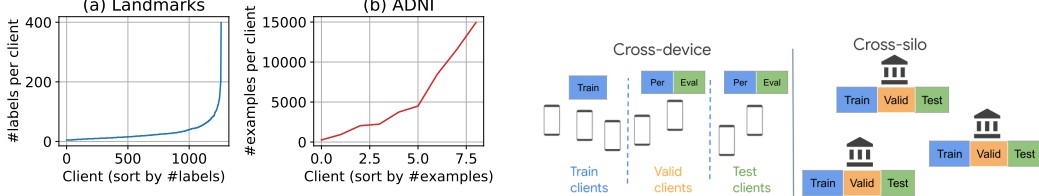

**Figure 1:** **Left**: The chosen federated datasets (Table 2) have heterogeneous local distributions (see Appendix E for other datasets). **Right**: To best reflect real-world FL applications, we naturally pre-process cross-device and cross-silo datasets differently: in cross-device, we split the clients into train/valid/test; in cross-silo, we split each client's local dataset (see Section 2.2).

---

[4][62] generalizes the original FedAvg algorithm [57] by using adaptive optimizers as the server optimizer.

## 2.2 Datasets and Pre-processing

Table 2 lists the datasets carefully chosen to reflect real-world FL applications, including a health data[5] and a school data[6] in the cross-silo setting (see Appendix E for detailed descriptions). All datasets have natural per-user partitions and distinct local statistics (Figure 1). Motley provides data pre-processsing pipelines for all data[1], and includes a critical and often overlooked distinction between pre-processing cross-device vs. cross-silo data. As shown in Figure 1, for **cross-device** data, we first randomly split the *clients* into three disjoint sets: train, validation (for hyperparameter tuning), and test (for final evaluation). This split reflects practical cross-device FL settings: given the population scale (e.g., millions of mobile devices [24]), devices participating in inference may never join training. Then we split each validation and test client's local examples into two equal-sized sets: a personalization set (for learning a personalized model) and an evaluation set (for evaluating the personalized model). For **cross-silo** data, because the total number of silos is small (e.g., tens of hospitals), and the same silos usually participate in both training *and* inference, we split each silo's local examples into three sets: a train, validation, and test set (Figure 1).

## 3   Cross-Device Experiments

The complete results on the cross-device datasets are in Appendix B (in particular, Table 4 has a detailed summary of the results). Next we briefly summarize the results and our findings.

As we focus on personalization, it is crucial to see how the per-client accuracy changes. As shown in Figure 2, fine-tuning improves the average per-client accuracy on EMNIST, StackOverflow, and Landmarks. Besides accuracy, we also report a fairness metric[7] in Table 4, and found that personalization algrorithms usually improve fairness.

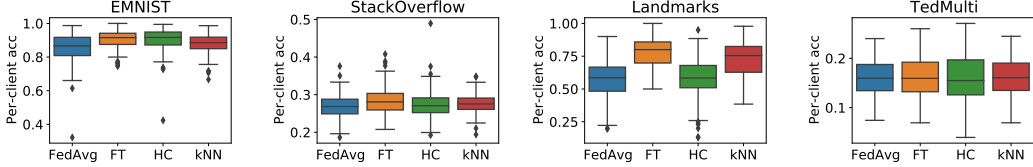

**Figure 2:** Test clients' per-client accuracy shown in box plots. Fine-tuning gives the best average per-client accuracy on EMNIST, StackOverflow, and Landmarks (more detailed results are in Table 4).

### 3.1   FedAvg + Fine-tuning (FT)

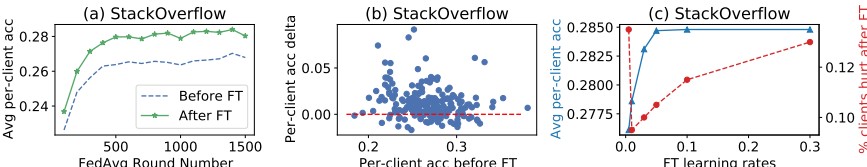

**Figure 3:** Fine-tuning may hurt some clients. (a) Fine-tuning improves average per-client accuracy. (b) Scatter plot of per-client accuracy delta (i.e., accuracy after fine-tuning – before fine-tuning). Each dot is a client. (c) The "avg per-client acc" metric may prefer a moderately large FT learning rate, while the "clients hurt" metric prefers a small FT learning rate.

**Observation 1: Fine-tuning can hurt clients.** As shown in Figure 3(b), some clients drop accuracy after fine-tuning (i.e., fall below the red dashed line at zero). There are two reasons: 1) The per-client accuracy metric is noisy due to small local data; 2) The fine-tuning hyperparameters are tune globally but different clients prefer different hyperparameters. See Appendix B.1 for the full explanation.

**Observation 2: Hyperparameter tuning can be difficult.** First, different metrics may favor different hyperparameters, as shown in Figure 3(c). Second, since the per-client accuracies are noisy, we need to compare the statistical significance of two results. Third, extra hyperparameters add another layer of complexity. For example, it may be helpful to tune FedAvg together with fine-tuning [35].

---

[5]Health data obtained from Alzheimer's Disease Neuroimaging Initiative: https://adni.loni.usc.edu/.
[6]Data collected by https://en.wikipedia.org/wiki/Inner_London_Education_Authority.
[7]There are many notions of fairness, e.g., group fairness [25]. In this paper we use a notion specific to federated learning (see, e.g., [49, 58]): all clients should have similar local model accuracies.

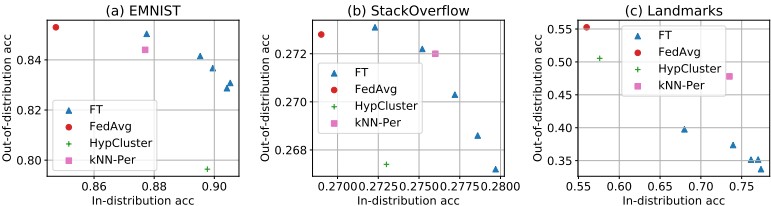

**Figure 4:** Trade-off between in-distribution (ID) and out-of-distribution (OOD) accuracy for different algorithms. We plot the results for fine-tuning epochs $1, 3, 5, 10, 15$. Increasing the fine-tuning epochs gives higher ID accuracy but lower OOD accuracy.

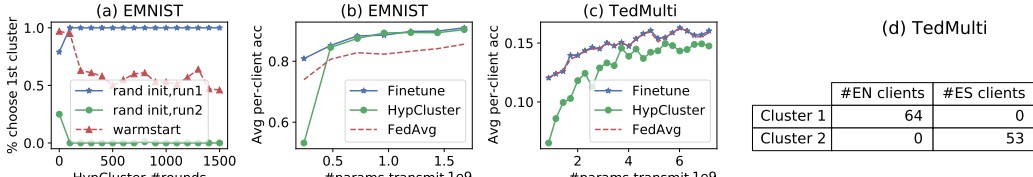

**Figure 5:** (a) Warmstart helps mitigate the mode collapse issue. (b) and (c) Plot of the average per-client accuracy as a function of communication cost for EMNIST and TedMulti. (d) Number of English and Spanish clients in the two clusters learned by HypCluster on TedMulti.

**Observation 3: Sensitivity to distribution shift.** Fine-tuning is able to quickly adapt a model to a client's local data, but is also prone to overfitting (a.k.a. catastrophic forgetting [56]). To illustrate this, we evaluate each test client's personalized model over the in-distribution (ID) test set (i.e., the client's local evaluation set[8]) and an out-of-distribution (OOD) test set. This OOD test set is formed by randomly sampling local examples from the test clients. This result is shown in Figure 4, which indicates an extra maintenance cost of personalized algorithms, i.e., decide when and how to continuously personalize the model when clients have new data.

**Remark: Acceptance criteria and robust personalization.** Our experiments assume that each client always uses the fine-tuned model for inference. In practice, one can design appropriate criteria so that a client only accepts the fine-tuned model under certain conditions [66]. Mitigating catastrophic forgetting [56] during model adaptation is an active research area, e.g., recent work [3, 33, 41, 77, 78], which is worth exploring in the personalized FL setting.

**Observation 4: Performance drops when clients have smaller personalization set[8].** As shown in Figure 6(c), where we reduce the number of examples in each test client's persoanlization set, the performance of fine-tuning and kNN-Per drops quickly. By contrast, HypCluster is quite robust to this change, which makes sense because HypCluster only needs the personalization set for model evaluation and choosing the best model, while fine-tuning and kNN-Per needs to train a new model.

### 3.2 HypCluster / IFCA

As shown in Figure 2, HypCluster performs worse than other two personalization algorithms.

**Observation 1: "Mode collapse" can hurt training.** "Mode collapse" happens when all clients will always choose the same cluster (Figure 5(a)). One way to mitigate this issue is to warm start HypCluster with models trained by FedAvg. However, mode collapse can still happen even with warm start. How to effectively train HypCluster without mode collapse remains an open problem.

**Observation 2: High communication cost per round.** A limitation of HypCluster is that in each round, the server needs to broadcast $k$ models (where $k$ is the number of learned clusters) to the clients, and hence, incurs $k$ times the communication cost of FedAvg[9]. This is shown in Figure 5(b-c).

**Observation 3: Difficulty in interpreting the learned clusters.** While it is generally hard to interpret the learned clusters on an arbitrary dataset, we do know that TedMulti-EnEs has two natural clusters of clients, i.e., English and Spanish users. Figure 5(d) shows that HypCluster indeed uncovers the two underlying clusters on TedMulti-EnEs.

---

[8]See Section 2.2, each validation and test client has two sets: a personalization set and an evaluation set.

[9]In real-world cross-device FL settings, reducing the communication cost between the server and mobile devices can help reduce the latency due to stragglers (see Section 5.1 in [74] for more discussions).

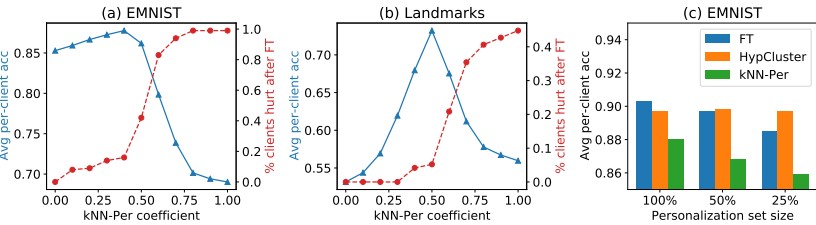

**Figure 6:** (a-b) Different metrics may be improved by different hyperpamater values: the "average per-client accuracy" metric usually prefers a moderately large interpolation coefficient while the "clients hurt" metric prefers a small coefficient. (c) Accuracy drops for fine-tuning and kNN-Per when each client has fewer examples to personalize the model. HypCluster is robust to this change because it only needs the personalization set for choosing the best model instead of training a model.

**Observation 4: Sensitivity to distribution shift.** We follow the same ID and OOD evaluation procedure in Section 3.1. As shown in Figure 4, HypCluster gives a worse ID-OOD tradeoff (i.e., achieves a lower OOD accuracy at the same ID accuracy) compared to the other algorithms.

### 3.3 FedAvg + kNN-Per

**Observation 1: Clients may be hurt.** This is illustrated in Figure 6(a-b). Section 3.1 gives two reasons why clients may hurt after fine-tuning, which also apply to kNN-Per.

**Observation 2: Difficulty in hyperparameter tuning.** The interpolation coefficient is a hyperparameter of kNN-Per. It is a scalar in (0, 1), where 0 means that no personalization and 1 means that the local prediction is formally entirely based on nearest neighbors. Figure 6(a-b) shows that different metrics may prefer different hyperparameter values. Section 3.1 points out two other reasons why hyperparameter tuning is difficult for fine-tuning. Both of them apply to kNN-Per as well.

**Observation 3: Sensitivity to distribution shift.** We follow the same ID and OOD evaluation procedure in Section 3.1. As shown in Figure 4, kNN-Per gives a similar or better ID-OOD tradeoff (i.e., achieves a higher OOD accuracy at the same ID accuracy) compared to the other algorithms.

**Observation 4: Performance drops when clients have smaller personalization set[8].** This has been discussed previously in Section 3.1.

## 4 Cross-Silo Experiments

We now consider baselines for the cross-silo personalization methods and datasets in `Motley`. The complete results are in Appendix C (see Table 5 for a detailed summary of the results).

**Results.** Most of the practical concerns of FedAvg+Fine-tuning and HypCluster discussed in Section 3 are still applicable here, e.g., HypCluster is still difficult to train due to the mode collapse issue (although it may happen less frequently when training simple linear models on Vehicle and School). In addition to the shared practical concerns, we observe four key trends specific to the cross-silo setting.

**Observation 1: Effectiveness of local training.** Local training may be a strong baseline in cross-silo settings. If we consider client(silo)-level joint differential privacy [34, 45], it has an extra benefit of no privacy cost.

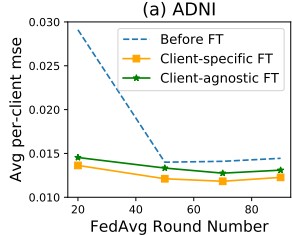

**Figure 7:** Tuning the fine-tuning hyperpameters in a per-client vs global manner.

**Observation 2: Importance of personalization.** The four personalization algorithms all achieve better mean accuracy (or MSE) and fairness[7] than that of vanilla FedAvg.

**Observation 3: Performance of MTL.** We see that (stateful) MTL methods could yield competitive performance with fine-tuning on the three datasets, with potentially less hyperparameter tuning.

**Observation 4: Effectiveness of client-specific hyperparameter tuning.** For FedAvg+Fine-tuning, we compared tuning the fine-tuning hyperparameters (i.e., fine-tuning learning rate and the number of fine-tuning epochs) in a per-client (every client chooses their own hyperparameter) vs global (all clients share the same hyperparameter) manner. Using client-specific fine-tuning hyperparameters can be better than tuning globally when each client's local data is large (see Figure 7).

# 5 Discussion and Open Directions

In this work we present `Motley`, a large-scale benchmark for personalized FL covering both cross-device and cross-silo settings. `Motley` provides a reproducible, end-to-end experimental pipeline including data preprocessing, algorithms, evaluation metrics, and tuned hyperparameters. Beyond these baselines, our experiments provide several new insights about personalized FL (see the detailed summary in Appendix H). These insights suggest several directions of future work, such as:

- The notion of the "best" method (or "best" hyperparameter of the same method, e.g., Figure 3) can change depending on the evaluation metric or setting (Table 4 and 5). A critical direction is thus to develop systematic evaluation schemes for personalized FL (i.e., mean accuracy alone is not enough).

- Existing literature often overlook or obfuscate the practical complexities of deploying personalized FL algorithms in real-world settings (e.g, Section 3.1 discusses the hyperparameter tuning difficulties due to local data scarcity and heterogeneity in cross-device FL). Designing new practical personalized FL methods that take these considerations into account is an important direction to democratize FL.

- Developing techniques to train and interpret clustering methods such as HypCluster without the "mode collapse" issue (see Figure 5 and Section 3.2) is a necessary step to make these approaches more effective in practice.

- Tradeoffs exists between adapting a client's personalized model to the current local distribution and generalizing to future distributions (Figure 4), which is worth exploring in greater detail (see the "Remark: Acceptance criteria and robust personalization" in Section 3.1).

- Given the observed benefits of per-client hyperparameter tuning in cross-silo FL (Figure 7), it may be beneficial to develop similar, scalable approaches for hyperparameter tuning in cross-device FL.

Finally, we note that the area of benchmarking itself can be improved in future iterations. For example, we hope that `Motley` can inspire benchmarking of additional evaluation metrics such as privacy, other notions of fairness[7], and robustness, and additional datasets and applications.

## Acknowledgements

We are grateful to Zachary Garrett, Jakub Konečnỳ, H. Brendan McMahan, Sewoong Oh, Daniel Ramage, Keith Rush, and Ananda Theertha Suresh for helpful discussions and comments.

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

# A `Motley`: A Benchmark for Personalized Federated Learning

Table 2 gives an overview of the three components of `Motley` (methods, datasets, baselines). We describe the included methods and datasets in this section. In Sections B and C, we provide baseline results and discuss the practical concerns of performing personalization in real-world FL applications.

## A.1 Personalization Methods

`Motley` includes five simple and model-agnostic algorithms for learning personalized models. The first four algorithms are *stateless*, and hence, are appropriate in the cross-device FL setting (see Section 1 and [36, 74] on why stateless algorithms are typically necessary for cross-device FL).

- **Local training** refers to every client training a local model using their own data, without collaborating with others. While it is not an FL algorithm, we include it in `Motley` as it provides a good baseline for FL algorithms, and it may be competitive when clients have sufficient local data [79].

- **FedAvg+Fine-tuning** [35] is a simple method for stateless, model-agnostic personalized FL, operating as follows: First train a global model via FedAvg [62][4]; then, each client fine-tunes the global model on their local data and uses the fine-tuned model for inference. It has a natural connection to meta-learning [35, 38][10], and has been reported to work well in real-world on-device applications [66, 75]. In `Motley`, we explore two variations: fine-tuning the entire model or fine-tuning only the last layer (the latter can be viewed as a form of hard parameter sharing MTL).

- **HypCluster** [53] (also **IFCA** [20]) is a stateless method that jointly clusters clients and learns a model for each cluster. Both HypCluster and IFCA work as follows: in each training round, the server sends all models (one per cluster) to the participating clients; each client receives the models and picks the cluster associated to the model with the lowest loss on its local data. It then computes an update for the selected model and sends the update and cluster identity to the server. The server aggregates the model updates for each cluster in the same way as in FedAvg [62][4]. In `Motley`, we explore *two initialization strategies*: random and warm start with FedAvg[4].

- **FedAvg+kNN-Per** [54] is a very recent personalization algorithm. The idea is to interpolate/ensemble the output of two models: a globally-trained FedAvg model and a local kNN model. For each client's local example, a representation vector is obtained from the FedAvg model (e.g., the input to the last softmax layer, or the states of an LSTM model). The Euclidean distance between the representations are used to learn a kNN model.

- **Multi-Task Learning** (MTL) is a class of methods used to deliver personalized models for a set of tasks by learning the task relations (either explicitly or implicitly). Each 'task' corresponds to a client in the FL setting. MTL approaches usually require the clients to be stateful [e.g., 21, 68], and hence, are more appropriate for cross-silo settings (see Section 1 on stateless vs stateful algorithms). Many existing personalized FL methods can be viewed as a form of MTL [e.g., 21, 22, 49, 68, 69]. In `Motley`, we consider two MTL algorithms: 1) Mocha [68], the first work that proposes to personalize federated models in convex settings, and 2) Ditto [49], a recent work that regularizes personalized models towards optimal global models for both convex and non-convex problems.

## A.2 Datasets and Pre-processing

Table 2 lists the datasets carefully chosen to reflect real-world FL applications (see Appendix E for detailed descriptions). All datasets have natural per-user partitions and distinct local statistics (Fig. 1). `Motley` provides data pre-processsing pipelines for all data[1], and includes a critical and often overlooked distinction between pre-processing cross-device vs. cross-silo data (described below).

**Pre-processing cross-device datasets.** Two steps are necessary to evaluate personalization algorithms in the cross-device setting. First, we randomly split the *clients* into three disjoint sets: train, validation (for hyperparameter tuning), and test (for final evaluation). This split reflects practical cross-device FL settings: given the population scale (e.g., millions of mobile devices [24]), devices participating in training are usually different from those in inference. Second, we split each validation and test client's local examples into two equal-sized sets: a personalization set and an evaluation set.

---

[10]Specifically, FedAvg can be viewed as performing the Reptile algorithm [60] to learn an effective starting model, such that after fine-tuning, the model can quickly adapt to a client's local data.

See Section B for how they are used in each algorithm. Specifically, the personalization set is used to train a local model, fine-tune a FedAvg-trained model, or select the best HypCluster model. On StackOverflow, we *sort* the local examples by time before splitting[11]. We perform random split on other datasets because they do not save time information[12].

**Pre-processing cross-silo datasets.** Unlike cross-device FL, in the cross-silo FL setting the total number of silos is small (e.g., tens of hospitals), and the same silos usually participate in both training *and* inference. To evaluate this setting, we split each silo's local examples into three sets: a train, validation, and test set. See Section C for details on how they are used in training and evaluation.

**Clients sampling.** Besides the distinction in the pre-processing steps, another distinction is the clients sampling rate during training. As mentioned in Section 1, the clients sampling rate is usually very low [36] in the cross-device setting, e.g., the total population is hundreds of millions devices but only a few thousands participate at every training round. Table 3 lists the number of clients sampled per round used in our cross-device experiments. In the cross-silo experiments, we assume that all clients are always available at each training round.

**Table 3:** Number of clients sampled per round in our cross-device experiments (see Appendix F for all the hyperparameters). The real-world cross-devices settings usually have a very low sampling rate, e.g., hundreds of millions devices in total but only a few thousands participate each round [36].

| Datasets | Total train clients | Sampled clients/round | Sampling rate |
|---|---|---|---|
| EMNIST | 2500 | 50 | 2% |
| StackOverflow | 342,477 | 200 | 0.05% |
| Landmarks | 1112 | 64 | 6% |
| TedMulti-EnEs | 3969 | 32 | 0.8% |

# B    Cross-Device Experiments

We now provide cross-device personalization baselines (Table 4) by evaluating the stateless methods discussed in Section A on the cross-device datasets included in `Motley`, and discuss our findings.

**Training and evaluation process.** As discussed in Section A.2, each cross-device dataset has train, validation, and test clients. We use the train clients to train a single model via FedAvg or a model per cluster via HypCluster, then *evaluate* the trained model(s) on the validation clients. For FedAvg + Fine-tuning, this evaluation is done on each client by first fine-tuning the FedAvg-trained model on the local personalization set, then evaluating the fine-tuned model on the local evaluation set. For HypCluster, this evaluation is performed on each client by using the local personalization set to find the model with the lowest loss, and then evaluating the selected model on the evaluation set. For FedAvg+kNN-Per, each client first uses the FedAvg-trained model to extract the example representations on the local personalization set, and use them to get a kNN model, and after that, evaluates the personalized model (interpolated between the kNN model and FedAvg-trained model) on the evaluation set. We use the validation metrics (i.e., the evaluation metrics over the validation clients) to select the best hyperparameters (see Appendix F) and report the test metrics (i.e., evaluation metrics on the test clients) in Table 4. We discuss additional details and results for each stateless FL method below.

## B.1    FedAvg + Fine-tuning (FT)

As we focus on personalization, it is crucial to see how the per-client accuracy changes with and without personalization. This is shown by the "Per-client acc before/after FT" metrics in Table 4, where we list the mean $\pm$ standard deviation of the test clients' per-client accuracy before and after fine-tuning. The mean and standard deviation metrics here are averaged over 5 runs[13]. As

---

[11]The model is fine-tuned on the old examples and evaluated on the new ones, reflecting practical FL settings.
[12]This means that, except StackOverflow, our experiments do not capture the potential distribution shift between the old and new examples. See Section B.1 for why this can potentially benefit FedAvg+Fine-tuning.
[13]See Appendix G for the standard deviations across the 5 runs.

**Table 4:** Summary of experimental results on the cross-device FL datasets as well as possible concerns around using each method in practice. We report the per-client accuracy (mean $\pm$ standard deviation) on the test clients after training 1500 rounds (following [14]) on EMNIST, StackOverflow and TenMulti, and 30k rounds (following [30]) on Landmarks. Note that the standard deviation is across all the clients' local accuracies, which is a fairness metric considered in [49, 58]. Each value is further averaged over 5 different runs (see Table 6 for the standard deviation over the 5 runs). Appendix F provides the tuned hyperparameters.

| Algorithm | Metrics | EMNIST | StackOverflow | Landmarks | TedMulti |
|---|---|---|---|---|---|
| Local training | Per-client acc | 0.594±.17 | 0.062±.03 | 0.173±.16 | 0.056±.02 |
| FedAvg + Fine-tuning (FT) | Per-client acc before FT | 0.844±.10 | 0.269±.03 | 0.564±.16 | 0.160±.04 |
| | Per-client acc after FT | 0.903±.06 | 0.282±.03 | 0.773±.11 | 0.162±.04 |
| | % clients "hurt" after FT | 5.2% | 14% | 5.6% | 40% |
| | FT all layers vs last layer | All layers | All layers | All layers | All layers |
| | *Practical concerns: difficult to tune hyperparameters; may hurt clients; sensitive to distribution shift; performance drops with fewer examples (see Section B.1)* | | | | |
| HypCluster / IFCA | Per-client acc | 0.897±.08 | 0.273±.03 | 0.573±.16 | 0.163±.04 |
| | No. tuned clusters ($k$) | 2 | 2 | 2 | 2 |
| | % clients largest cluster | 52.6% | 85.1% | 92.1% | 54.7% |
| | Warmstart from FedAvg | Yes | Yes | Yes | Yes |
| | Per-client acc by ensembling $k$ FedAvg models | 0.860±.08 | 0.271±.03 | 0.564±.16 | 0.163±.04 |
| | *Practical concerns: hard to train due to mode collapse; high communication cost; hard to interpret the clusters; sensitive to distribution shift (see Section B.2)* | | | | |
| FedAvg + kNN-Per | Per-client acc | 0.876±.06 | 0.275±.03 | 0.735±.13 | 0.162±.05 |
| | % clients "hurt" | 19.8% | 23.6% | 5.6% | 34.4% |
| | *Practical concerns: difficult to tune hyperparameters; may hurt clients; sensitive to distribution shift; performance drops with fewer examples (see Section B.3)* | | | | |

shown in Table 4 and Figure 2, fine-tuning improves the average per-client accuracy on EMNIST, StackOverflow, and Landmarks, and reduces the standard deviation across the clients' local model accuracies on EMNIST and Landmarks (i.e., improves *fairness*[7]). We explored fine-tuning the entire model or only last layer and found that fine-tuning all layers perform better, as shown by the metric "FT all layers vs last layer" in Table 4.

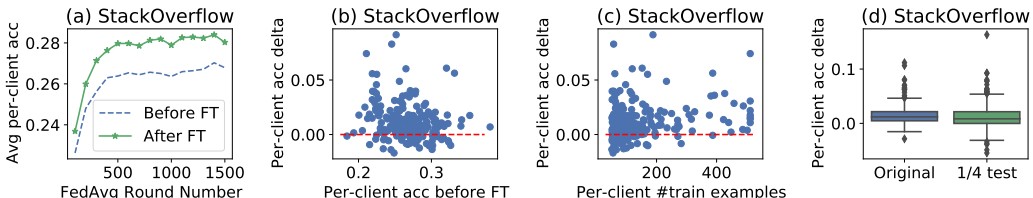

**Figure 8:** Fine-tuning may hurt some clients. (a) Fine-tuning improves average per-client accuracy. (b) Scatter plot of per-client accuracy delta (i.e., accuracy after fine-tuning – before fine-tuning). Each dot is a client. (c) Same scatter plot as (b) with the x-axis being the number of local examples used to fine-tune the model. (d) Fine-tune the same model on the same set of clients using smaller test data.

**Observation 1: Fine-tuning can hurt clients.** The metric "% clients hurt after FT" in Table 4 is the percentage of test clients whose local accuracy drops after fine-tuning. This phenomenon can be better visualized in Figure 8: while Figure 8(a) shows that fine-tuning increases the average per-client accuracy on the StackOverflow dataset, the scatter plot (each dot is a client) of the per-client accuracy deltas (i.e., accuracy after fine-tuning – before fine-tuning) in Figure 8(b) shows that some clients drop accuracy after fine-tuning (i.e., fall below the red dashed line at zero). We identify two reasons:

a. The per-client accuracy is noisy for clients with small local datasets. This effect occurs in two distinct ways: 1) Clients use a small number of examples to fine-tune the model[14]. In Figure 8(c), we show the same scatter plot (each dot is a client) as in Figure 8(b) with the x-axis being the number of examples used to fine-tune the model. The clients hurt by fine-tuning tend to have a small local personalization set[8]. 2) Clients use a small number of examples to evaluate the fine-tuned model. In Figure 8(d), we fine-tune the same model on the same set of clients. The only difference is the size of the evaluation set[8]. The evaluation metrics are noisier (i.e., we see a larger range in Figure 8(d)) for smaller evaluation sets.

b. Heterogeneity among clients and the fact that the fine-tuning hyperparameters are chosen globally. Since each client's local dataset is typically small, instead of choosing the fine-tuning hyperparameters (i.e., the fine-tuning learning rate, the number of fine-tuning epochs, and which layer to fine-tune) in a per-client manner[15], we tune them globally and apply the same hyperparameters to all test clients. The global fine-tuning hyperparameters work for most clients, but can adversely affect others. This also illustrates why hyperparameter tuning is difficult (see below).

**Observation 2: Hyperparameter tuning can be difficult.** At least three hyperparameters are specific to fine-tuning: which layers to fine-tune, the fine-tuning learning rate[16], and the number of epochs to fine-tune the model. As pointed out in Observation 1b), due the small local dataset on each client, instead of choosing the best fine-tuning hyperparameters in a per-client manner, we typically tune them globally and apply the same hyperparameters to all the test clients. Choosing a good set of fine-tuning hyperparameters can be particularly difficult due of the following reasons:

a. Different metrics may favor different hyperparameters[17]. For example, if we look at Figure 9(a), we may want to choose 0.1 as the fine-tuning learning rate because it gives higher average per-client accuracy; however, if we look at Figure 9(b), we may want to choose 0.25 because fewer clients drop their accuracies after fine-tuning (i.e., per-client acc delta $< 0$). In Figure 9(c), we plot the two metrics across different fine-tuning learning rates, and it is clear that different metrics may prefer different learning rates.

b. Since the per-client accuracies are noisy, we need to be careful when determining whether the difference between two sets of hyperparameters is statistically significant.

c. Extra hyperparameters add another layer of complexity. For example, it may be helpful to tune FedAvg together with fine-tuning so that the model produced by FedAvg is a good initial model for fine-tuning [35]. Existing FL hyperparameter optimization methods (e.g., [39]) often overlook the practical issues (e.g., clients may be hurt by fine-tuning, inconsistent and noisy metrics) mentioned above. Another example of an extra hyperparameter is that one may want to learn a threshold[14] such that only clients whose local dataset size is larger than this threshold can perform fine-tuning (see also the remark on "acceptance criteria and robust personalization" after Observation 3).

**Observation 3: Sensitivity to distribution shift.** Fine-tuning is able to quickly adapt a model to a client's local data, and hence, can achieve good test performance as shown in Table 2, especially on the two image datasets EMNIST and Landmarks. However, a potential drawback of this quick adaption is that the fine-tuned model may perform poorly if a client's future data distribution differs from the existing data (a.k.a. catastrophic forgetting [56]). To illustrate this, we evaluate each test client's personalized model over the in-distribution (ID) test set (i.e., the client's local evaluation set[8]) and an out-of-distribution (OOD) test set. This OOD test set is formed by randomly sampling local examples from the test clients. The averaged ID and OOD test accuracies are shown in Figure 4. We plot the results for different fine-tuning epochs. Larger fine-tuning epoch gives higher ID accuracy but lower OOD accuracy. This trade-off indicates an extra maintenance cost of personalized algorithms, i.e., decide when and how to continuously personalize the model when clients have new data.

**Remark: Acceptance criteria and robust personalization.** Our experiments assume that each client always performs fine-tuning and uses the fine-tuned model for inference. In practice, one can design appropriate criteria so that a client only performs fine-tuning and accepts the fine-tuned model under certain conditions. For example, a client can perform fine-tuning only when its local dataset

---

[14]In practice, it may be good to only apply fine-tuning to clients with large amount of local data.

[15]We tried this approach (i.e., each client chooses the fine-tuning hyperparameters based on its local data) on the StackOverflow dataset, but found that it performed worse than tuning the hyperparameters globally.

[16]We focus on SGD when fine-tuning the model, as adaptive optimizers seem to perform similarly as SGD.

[17]The results shown in Table 4 is based on tuning hyperparameters by the average per-client accuracy.

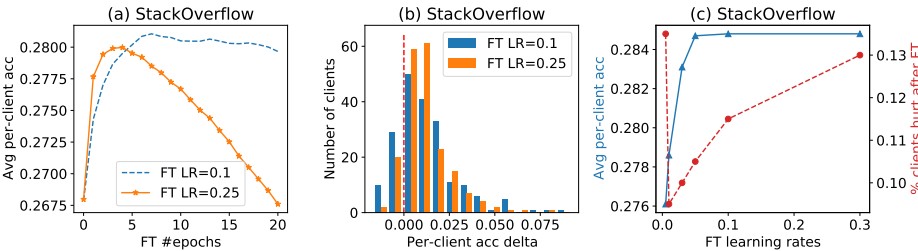

**Figure 9:** Choosing a good set of fine-tuning hyperparameters is difficult, as different metrics may be improved by different hyperparameters. (a) FT learning rate 0.1 gives higher average per-client accuracies; (b) FT learning rate 0.25 hurts fewer clients; (c) The "avg per-client acc" metric may prefer a moderately large FT learning rate, while the "clients hurt" metric prefers a small FT learning rate.

size is large enough as mentioned previously[14]; a client can only accepts the fine-tuned model when its evaluation metric is better than that of the baseline model. See Section 2.2 of [66] for an example of acceptance criteria developed for a real-world on-device personalization application. Designing a good set of acceptance criteria can mitigate the negative effects due to Observation 1 and 3 above, and potentially prevent clients from accepting fine-tuned models that do not generalize well. Mitigating catastrophic forgetting [56] during model adaptation is an active research area, e.g., some recent work [3, 33, 41, 77, 78], which would be worth exploring in the setting of personalized federated learning.

**Observation 4: Performance drops when clients have smaller personalization set[8].** This is already indicated previously in Figure 8(c), which shows that clients with smaller personalization set (i.e., fewer examples used to fine-tuned the model) tend to be hurt by fine-tuning. In Figure 6(c), we reduce the number of examples in each test client's persoanlization set to 50% and 25% of its original size, and report the accuracy averaged over the test clients. We see that the performance of fine-tuning and kNN-Per drops quickly when we decrease the number of local examples to personalize the model. On the other hand, HypCluster is quite robust to this change, which makes sense because HypCluster only needs the personalization set for model evaluation and use the evaluation metrics to choose the best model, while fine-tuning and kNN-Per needs the local examples to train a new model.

### B.2 HypCluster / IFCA

We now discuss results of HypCluster/IFCA algorithm. As shown in Table 2, the average per-client accuracy of HypCluster is slightly better than that of FedAvg[18], but is much worse than that of FedAvg+Fine-tuning on the first three datasets. We tuned[19] the number of clusters between {2,3,4}; however, due to the mode collapse issue (described below), we found that it difficult to learn $k$ models with $k > 2$. A natural baseline to compare with HypCluster is to run FedAvg $k$ times, and ensemble[20] the $k$ models. We also include this baseline in Table 2, and discuss it in Observation 3 below.

**Observation 1: "Mode collapse" can hurt training.** As shown in Figure 5(a), we monitor the percentage of clients that choose the first cluster when running HypCluster with random initialization on EMNIST. After a few rounds, all clients will always choose the same cluster. The same thing happens to all the other datasets. One way to mitigate this "mode collapse" issue is to warm start HypCluster with models trained by FedAvg. In our experiments, we train a few rounds of FedAvg for $k$ times, and use the $k$ models to warm start HypCluster (see Appendix F for the hyperparameters.). Figure 5(a) shows that warm start can help mitigate the mode collapse issue on EMNIST. However, mode collapse can still happen even when warm start is used, especially for $k > 2$. How to effectively train HypCluster without mode collapse in real-world cross-device settings remains an open problem.

---

[18]We follow the same training and evaluation process described in the beginning of Section B, and hence, the per-client accuracies of HypCluster shown in Table 2, can be directly compared to those of FedAvg+Fine-tuning.

[19]Except for TedMulti-EnEs which we focus on 2 clusters since it contains two languages: English/Spanish.

[20]There are many ways to combine multiple models, e.g., average the model outputs. Here we use the same procedure as in HypCluster, where each client selects the model with the lowest local loss for inference.

**Observation 2: High communication cost per round.** A limitation of HypCluster is that in each round, the server needs to broadcast $k$ models (where $k$ is the number of learned clusters) to the clients, and hence, incurs $k$ times the communication cost of FedAvg[21]. In Figure 5(b) and (c), we plot the average per-client accuracies with respect to the total number of parameters communicated (including those used for training the warm start models) for EMNIST and TedMulti. While TedMulti eventually achieves a slightly better average accuracy than FedAvg (Table 2), it is arguably worse than FedAvg from a accuracy-communication ratio perspective. While one can reduce communication costs via weight-sharing [20], we found that this increased the likelihood of mode collapse[22]. In future work it would be interesting to explore connections between mode collapse and weight sharing, and consider how to best train HypCluster in constrained networks.

**Observation 3: Difficulty in interpreting the learned clusters.** As discussed in the beginning of Section B.2, a natural baseline to compare with HypCluster is to run FedAvg $k$ times, and ensemble[20] the $k$ models. Specifically, we train HypCluster with $k$ clusters for $y$ rounds, and compare it with training FedAvg $k$ times and each for $y$ rounds, so both have the similar communication cost. As shown in Table 2, the average per-client accuracy of ensembling $k$ FedAvg models is similar to that of HypCluster for 3/4 datasets, so a natural question is: does HypCluster actually learn a meaningful cluster structure of the underlying data? While it is generally hard to answer this question on an arbitrary dataset, we do know that TedMulti-EnEs has two natural clusters of clients, i.e., English and Spanish users. Figure 5(d) lists the number of English and Spanish clients (total 117 test clients) in the two learned clusters. HypCluster indeed uncovers the two underlying clusters on TedMulti-EnEs.

**Observation 4: Sensitive to distribution shift.** We follow the same ID and OOD evaluation procedure in Section B.1. As shown in Figure 4, HypCluster seems to give a worse ID-OOD tradeoff (i.e., achieves a lower OOD accuracy at the same ID accuracy) compared to the other personalization algorithms.

### B.3   FedAvg + kNN-Per

We now discuss the results of kNN-Per, which are shown in Table 4 as well as in Figure 2.

**Observation 1: Clients may be hurt.** In Table 4, besides the per-client accuracy, we also list the percentage of clients "hurt" after personalization (i.e., test clients whose accuracy on the local evaluation set[8] after kNN-Per is lower than that of FedAvg). As shown in Table 4, compared to fine-tuning, kNN-Per has a lower "per-client acc" but a higher "% clients hurt" for 3/4 datasets. The two reasons why clients may hurt after fine-tuning given in Section B.1 also applies to kNN-Per: 1) The per-client accuracy metric is noisy due to local data scarcity; 2) The hyperparameter is chosen globally but each client has heterogeneous local distribution. Similar to fine-tuning, so far we assume all test clients will perform kNN-Per. Designing a good set of conditions as to when a client should perform kNN-Per may help reduce the fraction of client hurt (see "Remark: Acceptance criteria and robust personalization" in Section B.1).

**Observation 2: Difficulty in hyperparameter tuning.** Following [54], we set $k$ (the number of nearest neighbors) to 10 for all kNN-Per experiments ([54] shows that kNN-Per is robust to the choice of $k$). We tune the interpolation coefficient globally, i.e., all clients use the same coefficient[23]. The interpolation coefficient is a scalar in (0, 1) (see Eq.(7) of [54]), where 0 means that no personalization and 1 means that the local prediction is formally entirely based on nearest neighbors. Figure 6(a-b) plot the two metrics "average per-client accuracy" and "% client hurt" for different interpolation coefficients. We see that the two metrics may prefer different hyperparameter values. Section B.1 points out two other reasons why hyperparameter tuning is difficult for fine-tuning. Both of them apply to kNN-Per as well, including the noisy local metrics and potentially extra hyperparameters.

**Observation 3: Sensitive to distribution shift.** We follow the same ID and OOD evaluation procedure in Section B.1. As shown in Figure 4, kNN-Per seems to give a similar or better ID-OOD tradeoff (i.e., achieves a higher OOD accuracy at the same ID accuracy) compared to the other

---

[21]In real-world cross-device FL settings, reducing the communication cost between the server and mobile devices can help reduce the latency due to stragglers (see Section 5.1 in [74] for more discussions).

[22]One possible reason is that during training, the shared layers may produce features specifically tied to one cluster. To avoid this correlation, one may want to learn invariant features [5] that are good for all clusters.

[23]We also tried tuning a per-client specific coefficient but found this performed worse than global tuning due to limited amount of local data.

**Table 5:** Summary of experimental results on the cross-silo datasets. Similar to Table 4, we report the per-client test metric (mean $\pm$ standard deviation) *across the clients*, so the standard deviation can be viewed as a fairness metric [49, 53]. Each metric value is averaged over 5 independent runs with different random seeds (see Table 7 for standard deviations over the 5 runs). Vehicle uses accuracy as the metric while ADNI and School use mean squared error (MSE). We train linear models on Vehicle and School. Appendix F has the tuned hyperparameters.

| Algorithm | Metrics | Vehicle (acc) | School (mse) | ADNI (mse) |
|---|---|---|---|---|
| Local training | Per-client metric | 0.9367±.0248 | 0.0121±.0059 | 0.0177±.0106 |
| FedAvg + Fine-tuning (FT) | Per-client metric before FT | 0.8859±.0833 | 0.0130±.0068 | 0.0141±.0090 |
| | Per-client metric after FT | 0.9385±.0253 | 0.0116±.0056 | 0.0124±.0082 |
| | % clients "hurt" after FT | 4.4% | 33.0% | 0 |
| | FT all layers vs last layer | N/A | N/A | no difference |
| | *Practical concerns: Similar to those in the cross-device setting (Table 4); Tuning per-client hyperparameter may outperform global tuning (Fig. 7).* | | | |
| HypCluster / IFCA | Per-client metric | 0.9246±.0288 | 0.0112±.0053 | 0.0137±.0093 |
| | No. tuned clusters ($k$) | 4 | 3 | 2 |
| | % clients largest cluster | 49.6% | 44.6% | 78% |
| | Warmstart from FedAvg | No | Yes | Yes |
| | Per-client metric by ensembling $k$ FedAvg models | 0.8851±.0828 | 0.0129±.0066 | 0.0133±.0091 |
| | *Practical concerns: Similar to the cross-device setting (Table 4); on Vehicle and School, mode collapse occurs less potentially from using a linear model.* | | | |
| FedAvg + kNN-Per | Per-client metric | 0.9228±.0287 | 0.01163±.0055 | 0.0126±.0096 |
| | % clients "hurt" | 22.6% | 37.4% | 37.8% |
| | *Practical concerns: Similar to those in the cross-device setting (Table 4).* | | | |
| MTL (Ditto) | Per-client metric | 0.9377±.0218 | 0.0114±.0053 | 0.0134±.0063 |
| MTL (Mocha) | Per-client metric | 0.9371±.0244 | 0.0121±.0059 | N/A |

personalization algorithms. Nevertheless, it is still worth exploring methods that improve the OOD robustness during personalization, as discussed in the "Remark: Acceptance criteria and robust personalization" in Section B.1.

**Observation 4: Performance drops when clients have smaller personalization set[8].** This has been discussed previously in Section B.1. As shown in Figure 6, the performance of kNN-Per and fine-tuning drops quickly when the test clients have fewer examples to personalize the model (in the case of kNN-Per, fewer examples to find the neareast neighbors). This is in contrast to HypCluster, which only uses the local personalization set for identifying the best model instead of training a new model as in fine-tuning and kNN-Per.

## C  Cross-Silo Experiments

We now consider baselines for the cross-silo personalization methods and datasets in `Motley`.

**Training and evaluation setup.** As mentioned in Section A.2, in the cross-silo setting, the same silos appear in both training and inference; to evaluate this setting, we split each client's local examples into three sets: train/validation/test. We train on the training sets across all clients, tune hyperparameters on the validation sets (see Appendix F.3), and report metrics on the test sets. Unlike the cross-device setting, in the cross-silo setting, each client typically has sufficient compute power and a large number of local examples. This allows us to tune fine-tuning hyperparameters in a per-client manner (e.g., every client chooses their own hyperparameters based on their validation metric), as opposed to tuning globally in the cross-device setting in Section B.1.

**Results.** Table 5 reports the cross-silo experimental results in the same format as in Table 4. Most of the practical concerns of FedAvg+Fine-tuning and HypCluster discussed in Section B are still

applicable here, e.g., HypCluster is still difficult to train due to the mode collapse issue (although it may happen less frequently when training simple linear models on Vehicle and School). In addition to the shared practical concerns, we observe four key trends specific to the cross-silo setting.

**Observation 1: Effectiveness of local training.** Local training may be a strong baseline in cross-silo settings. If we consider client(silo)-level differential privacy, it has an extra benefit of no privacy cost.

**Observation 2: Importance of personalization.** The three personalization algorithms (FedAvg + Fine-tuning, HypCluster, and MTL) all achieve better mean accuracy (or MSE) and fairness[7] (i.e., smaller per-client metric standard deviation in Table 5) than that of FedAvg.

**Observation 3: Performance of MTL.** We see that (stateful) MTL methods could yield competitive performance with finetuning on the three datasets, with potentially less hyperparameter tuning.

**Observation 4: Effectiveness of client-specific fine-tuning.** For FedAvg+Fine-tuning, we compared tuning the fine-tuning hyperparameters (i.e., fine-tuning learning rate and the number of fine-tuning epochs) in a per-client (every client chooses their own hyperparameter) vs global (all clients share the same hyperparameter) manner. Using client-specific fine-tuning hyperparameters can be better than tuning globally when each client's local data is large (see the ADNI result in Figure 7).

## D   Background on Personalized Federated Learning

Traditionally, federated learning objectives consider fitting a single global model, $w$, across all local data in the network. The aim is to solve:

$$\min_w G(F_1(w), \ldots F_K(w)), \tag{1}$$

where $F_k(w)$ is the local objective for client $k$, and $G(\cdot)$ is a function that aggregates the local objectives $\{F_k(w)\}_{k \in [K]}$ from each client. For example, in FedAvg [57], $G(\cdot)$ is typically set to be a weighted average of local losses, i.e., $\sum_{k=1}^K p_k F_k(w)$, where $p_k$ is a pre-defined non-negative weight such that $\sum_k p_k = 1$. However, in general, each client may generate data $x_k$ via a distinct distribution $\mathcal{D}_k$, i.e., $F_k(w) := \mathbb{E}_{x_k \sim \mathcal{D}_k}[f_k(w; x_k)]$. To better account for this heterogeneity, it is therefore increasingly common to consider techniques (described below) that learn personalized, client-specific models, $\{w_k\}_{k \in [K]}$ across the network.

A distinguishing factor of personalized FL methods is whether the approach requires any variables to be maintained on participating clients from one round to another [74]. As discussed in Section 1, whereas stateless approaches may be applicable to either cross-device or cross-silo FL, stateful approaches are more appropriate for cross-silo settings given the typical size and configuration of the network. Besides stateful vs stateless, the personalized FL algorithms can be categorized into model-agnostic and model-specific. Model-specific approaches target a specific model and usually require domain-specific information [34, 67, 84]. The current version of `Motley` focuses on benchmarking personalization algorithms that can work without assumptions on the model or application scenario. We discuss major model-agnostic approaches in stateful vs. stateless personalized FL below, and defer readers to the recent surveys [70][74, §7.5] for more related work on personalized FL.

**Stateful Approaches.** A common class of stateful personalized FL methods are *multi-task learning (MTL)* methods. These methods view each client (or group of clients) as a separate 'task', and aim to jointly learn task-specific models while exploiting similarities/differences between tasks. The idea of solving multiple learning tasks simultaneously was first popularized by Caruana [11] in the 90's, who described multi-task learning as a technique in which related tasks act as a form of inductive bias to improve generalization. Many approaches can be captured in the following general and widely-used formulation, known as multi-task relationship learning [68, 80, 81]:

$$\min_{\boldsymbol{W}, \boldsymbol{\Omega}} \left\{ \sum_{k=1}^K \sum_{i=1}^{n_k} \ell_k(\boldsymbol{w}_k; \boldsymbol{x}_k^i, y_k^i) + \mathcal{R}(\boldsymbol{W}, \boldsymbol{\Omega}) \right\}. \tag{2}$$

Here $\boldsymbol{W} := [\boldsymbol{w}_1, \ldots, \boldsymbol{w}_K] \in \mathbb{R}^{d \times K}$ is a matrix whose $k$-th column is the weight vector for the $k$-th task (in this case, the model on client $k$). The matrix $\boldsymbol{\Omega} \in \mathbb{R}^{K \times K}$ models relationships amongst tasks, and is either known a priori or estimated while simultaneously learning task models. MTL problems differ based on their assumptions on $\mathcal{R}$, which takes $\boldsymbol{\Omega}$ as input and promotes some

suitable structure amongst the tasks. In federated learning, a number of specific MTL instantiations have been proposed, including variants of general relationship learning [32, 68], cluster-regularized MTL [65, 68], global-regularized MTL [49, 79], and mean-regularized MTL [17, 21, 22].

Beyond these approaches, another common variant of MTL, particularly for deep learning problems, is 'hard parameter' sharing approaches [63]. These methods consider splitting the model architecture itself into two components: a shared part that is jointly learned by all clients, and a local part that is personalized to each client. The local portion can be simply fine-tuned (as discussed below), or may be learned in conjunction with the shared portion by saving the local state on each client at every round, in which case they would also fall under the category of stateful FL [4, 23, 52].

**Stateless Approaches.** In the stateless category, one of the most common forms of personalization is simple fine-tuning. With fine-tuning, a shared model is trained and deployed on each client, and the model is then fine-tuned or adapted locally to the client's data. In its simplest form the deployed model could be a global model trained via a typical procedure such as FedAvg [57, 62], and additional iterations of a stochastic optimizer such as mini-batch SGD can be run locally after deployment. However, it is also natural to consider *meta-learning* approaches as part of this workflow [13, 19, 35, 38], which are specifically designed to learn an algorithm that can solve a new task with a small number of training samples. These approaches can be particularly useful in cross-device FL settings, where each client may generate only a small number of training points, and it may be necessary to deploy and adapt models to clients that did not participate in training.

Finally, unlike the stateful MTL-based clustering approaches discussed above, it is also possible to consider clustering variants that don't require state to be maintained. In particular, a common approach is to maintain multiple global models and to have participating clients determine which of the models is best suited to their local data, thus forming a natural clustering amongst the clients [20, 53].

**In developing baselines for** `Motley`, we explore a simple fine-tuning after FedAvg approach (fine-tuning different layers), and a stateless clustering algorithm [20, 53] (different initialization schemes) for both cross-silo and cross-device settings. We also include a baseline of training local models without federation. Furthermore, we explore a stateful baseline of MTL (both simpler and more complex approaches) for the cross-silo setting. See Section A.1 for more details.

# E    Datasets

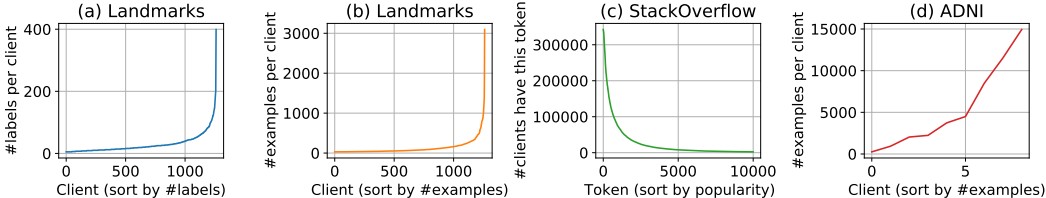

**Figure 10:** The chosen federated datasets (Table 2) have heterogeneous local distributions.

All the datasets used in `Motley` are listed in Table 2. They do not contain personally identifiable information or offensive content. They all have natural per-client data partitions, which will be discussed in more details below. Figure 10 shows that each client's local data distribution is distinct from each other, a common property of real-world federated datasets (see the discussions on "Heterogeneity" in Section 1 and Appendix D).

## E.1    Cross-device

We describe the cross-device datasets in Table 2. We will also mention how we split the clients into train, validation, and test clients (a key preprocessing step described in Section A.2).

- **EMNIST** is first processed by the LEAF benchmark [10]. We use the version from [6], which contains 3400 writers/clients and a total of 671,585 images[24]. Each client has on average 200 images (28-by-28 pixels) of hand-written digits and characters (62 classes). We use a CNN model [43] with 1M parameters (same as [14]). The 3400 clients are randomly split into 2500/400/500 as train/validation/test.

- **StackOverflow** is a large-scale federated language dataset. Each client is a user in the Stack Overflow online forum. The examples are the questions and answers posted by this user. Each example is a sentence. The task is next word prediction. We use an LSTM model [28] with 4M parameters and 10k vocabulary size (same as [14]). The version from [7] contains 342k train clients and 38k held-out clients. We randomly split the held-out clients into 10k/28k for validation/test.

- **Landmarks** is a federated image classification dataset processed by Hsu et al. [30] (from the 2019 Landmark Recognition Challenge [76]). Each client is a Wikipedia contributor. The images are the landmark (e.g., famous monuments and mountains) photos uploaded by the photographers. The per-user data distribution naturally varies based on the photographer's geographic location. The dataset contains 164k images, 2028 landmark labels, and 1262 clients. We use a MobileNetV2 model [64] with 4M parameters (same as [30]). We randomly split the clients into 1112/50/100 as train/validation/test.

- **TedMulti-EnEs** is a subset of the TedMultiTranslate dataset. The original TedMultiTranslate is a multilingual (60 languages) dataset derived from the TED Talk transrcipts [61]. We only use two languages English and Spanish (and hence, "EnEs" in the name). This dataset has not been used in any federated learning experiment before, so we need to do more preprocessing work. We first partition the clients by the TED Talk author and language (a client contains either English data or Spanish data but not both[25]). Each example is a sentence from the talk. The task is next word prediction. We build a vocabulary of size 15k containing both English and Spanish words (these words appear in at least 20 clients). We use a transformer model [71] with 4M parameters. Clients with less than 20 examples are removed. We randomly split the 4184 clients into 3969/98/117 as train/validation/test.

## E.2   Cross-silo

- **Vehicle** dataset, originally collected by Duarte and Hu [18], is a binary classification dataset containing measurements of road segments from a distributed network of 23 vehicle sensors with a total of 43,695 feature vectors for classifying the type of the passing vehicles. Following Smith et al. [68], we treat each sensor as a task (data silo) and use 50 acoustic and 50 seismic features as inputs to a linear SVM model. It is suitable for cross-silo FL because of the small number of silos and sufficiently large local datasets for local training. We split each client(silo)'s local dataset into 70%/15%/15% train/validation/test sets.

- **School** dataset, originally collected by the now-defunct Inner London Education Authority,[26] is a regression dataset for predicting the exam scores of 15,362 students distributed across 139 secondary schools. Each school has records up to 251 students with each student described by a 28-dimensional feature vector capturing attributions such as the school ranking, student birth year, and whether the school provided free meals. Simple linear regression models suffice for reasonable performance. We split each client(silo)'s local dataset into 70%/15%/15% train/validation/test sets.

- **ADNI** (Alzheimer's Disease Neuroimaging Initiative)[27] is a public medical dataset containing various formats of data to help advance the study of Alzheimer's disease.[28] The task is to predict Standardized Uptake Value Ratio (SUVR) from PET scans of human brains. We treat each

---

[24]Note that in the EMNIST dataset provided by [6], each client's examples are already split into a training set and a small test set, for simplicity, we only use the training set of each client in our experiments.

[25]The processed dataset has two clusters of clients: English/Spanish. See Section B.2 for more discussions.

[26]https://en.wikipedia.org/wiki/Inner_London_Education_Authority

[27]All research on the ADNI dataset was conducted by Tian Li. No other authors had access to this data. ADNI data were obtained from the Alzheimer's Disease Neuroimaging Initiative (ADNI) database (adni.loni.usc.edu). As such, the investigators within the ADNI contributed to the design and implementation of ADNI and/or provided data but did not participate in analysis or writing of this paper. A complete listing of ADNI investigators can be found at: http://adni.loni.usc.edu/wp-content/uploads/how_to_apply/ADNI_Acknowledgement_List.pdf. ADNI project was funded by National Institutes of Health Grant U01 AG024904 and DOD ADNI (Department of Defense award number W81XWH-12-2-0012).

[28]https://adni.loni.usc.edu/

scanner vendor as a silo, and there are 9 silos in total. We specifically use a subset of PET scans (with AV45 and preprocessing step 'Coreg, Avg, Std Img and Vox Siz, Uniform Resolution') that have existing labels obtained from UC Berkeley study in the database. We convert the data into .png format, normalize each pixel value to $(0, 1)$, and rescale each image into size $32 \times 32$. We split each client(silo)'s local dataset into 80%/10%/10% train/validation/test sets.

### E.3 License/Usage Information

- **EMNIST.** The Extended MNIST dataset[29] is a variant of the original NIST Special Database 19[30]. In this paper, we use a federated version of this Extended MNIST dataset, which is processed and provided by the LEAF[31] benchmark [10] under the BSD-2-Clause license.

- **StackOverflow.** This dataset is derived from the Stack Overflow Data hosted by Kaggle[32], under the CC BY-SA 3.0 license. In this paper, we use the federated version of this dataset processed by Tensorflow Federated [7], which is open-sourced under the Apache-2.0 license.

- **Landmarks.** The original Google Landmarks Dataset v2 [76] can be accessed from GitHub[33]: "The annotations are licensed by Google under CC BY 4.0 license. The images listed in this dataset are publicly available on the web, and may have different licenses". In this paper, we use a subset of the full dataset processed by Hsu et al. [30]: the data split function is available on GitHub[34] under the Apache-2.0 license.

- **TedMulti-EnEs.** The original TedMultiTranslate dataset is derived from the TED Talk transcripts [61]. The TED Talk usage policy can be found from their website[35]. The TedMultiTranslate dataset can be downloaded from GitHub[36]. We were unable to find the license information. In this paper, we extract a subset of this dataset and group the examples by talk author and language[25]. This preprocessing pipeline is available from our benchmark code[1] under the Apache-2.0 license.

- **Vehicle.** The Vehicle dataset is made publicly available by the original authors as a research dataset [18]. It has been subsequently used in recent work (e.g. [68]). License information was unavailable online. A copy of the dataset may be obtained from these URLs.[37,38]

- **School.** The School dataset was collected by a now-defunct entity and we were unable to find license information online. The dataset has been made freely available online and can be obtained from this URL.[39] Alternatively, a copy may also be obtained from [83] under the GNU GPL v2 license.

- **ADNI.** The term of use of ADNI can be found on their website.[40]

## F   Hyperparameters and Implementation Details

In this section, we describe the computation resources, the hyperparameter grids and the best hyperparameters for all the experiments described in Section B and Section C.

---

[29]https://www.westernsydney.edu.au/icns/reproducible_research/publication_support_materials/emnist
[30]https://www.nist.gov/srd/nist-special-database-19
[31]https://github.com/TalwalkarLab/leaf
[32]https://www.kaggle.com/stackoverflow/stackoverflow
[33]https://github.com/cvdfoundation/google-landmark
[34]https://github.com/google-research/google-research/tree/master/federated_vision_datasets
[35]https://www.ted.com/about/our-organization/our-policies-terms/ted-talks-usage-policy
[36]https://github.com/neulab/word-embeddings-for-nmt
[37]https://web.archive.org/web/20110515133717/http://www.ece.wisc.edu:80/~sensit/
[38]https://web.archive.org/web/20200128092656/http://www.ecs.umass.edu:80/~mduarte/Software.html
[39]https://web.archive.org/web/20060718012309/http://www.mlwin.com/intro/datasets.html
[40]https://adni.loni.usc.edu/terms-of-use/

### F.1 Cross-device

The definitions of all hyperparameters can be found in `finetuning_trainer.py` and `hypcluster_trainer.py` in our code, which will be open-sourced after the paper is accepted.

**Common hyparameters.** The following hyperparameters are fixed across all cross-device experiments. Note that we focus on *FedAdam* here, i.e., a generalized version of FedAvg [62], where the server optimizer is Adam optimizer [40], because [14] shows FedAdam gives good performance across different datasets.

- `client_optimizer=['sgd']`
- `server_optimizer=['adam']`
- `server_adam_beta_1=[0.9]`
- `server_adam_beta_2=[0.99]`
- `train_epochs=[1]`
  (This is the number of local training epochs performed by a client during a round of training.)

**Computation resources.** We summarize the computation resources allocated to run one experiment (i.e., one point in the the hyperparameter grids) on each dataset: EMNIST (80 CPU cores); StackOverflow (400 GPU cores); Landmarks (16 GPUs); TedMulti-EnEs (2 GPUs). Note that the actual usage may be smaller than the allocated resources.

#### F.1.1 FedAvg (i.e., FedAdam in our case) + Fine-tuning

The definitions of all the hyperparaemeters for running this algorithm can be found in `finetuning_trainer.py`. Since FedAvg + Fine-tuning is a two step process, the hyperparameters contain FedAvg (i.e., FedAdam in our case) hyperparameters and fine-tuning hyperparameters.

For EMNIST and StackOverflow, we use the best FedAdam hyperparameters from [14]. For Landmarks, we use the best FedAdam hyperparameters from [74]. For TedMulti-EnEs, we tune the FedAdam hyperparameters from scratch.

We have four fine-tuning hyperparameters. Their names are all started with `finetune_`, including the optimizer used to fine-tune the model (where we focus on SGD[16]), the fine-tuning learning rate, and whether to only fine-tune the last layer. We also need to tune the number of local epochs used to fine-tune the model - this value is automatically found by `finetuning_trainer.py` by postprocessing the validation metrics. Specifically, we compute the average validation accuracy of the fine-tuned models after every fine-tuning epoch (until `finetune_max_epochs`), and then find the best fine-tuning epoch that gives the highest average validation accuracy (see, e.g., Figure 9(a)[41]). The best fine-tuned epoch will be in the range [0, `finetune_max_epochs`], so all we need is to set a proper value for `finetune_max_epochs`.

**EMNIST**

Fixed hyperparameters (we use the best FedAdam hyperparameters from [14]):

- `client_learning_rate=[0.1]`
- `server_learning_rate=[0.001]`
- `server_adam_epsilon=[0.001]`
- `clients_per_train_round=[50]`
- `train_batch_size=[20]`
- `total_rounds=[1500]`
- `valid_clients_per_round=[100]`
- `test_clients_per_round=[100]`

---

[41]This figure shows how average test accuracy changes with respect to the fine-tuning epochs. Average validation accuracy follows a similar trend as the test accuracy.

- `rounds_per_evaluation=[100]`
- `rounds_per_checkpoint=[100]`
- `finetune_optimzier=['sgd']`
- `finetune_max_epochs=[20]`

Tuned hyperparameters (best values are highlighted in **value**):

- `finetune_learning_rate=[0.001, 0.003, **0.005**, 0.01, 0.05]`
- `finetune_last_layer=[True, **False**]`

### StackOverflow

Fixed hyperparameters (we use the best FedAdam hyperparameters from [14]):

- `client_learning_rate=[1.0]`
- `server_learning_rate=[0.1]`
- `server_adam_epsilon=[0.001]`
- `clients_per_train_round=[200]`
- `train_batch_size=[16]`
- `total_rounds=[1500]`
- `valid_clients_per_round=[200]`
- `test_clients_per_round=[200]`
- `rounds_per_evaluation=[100]`
- `rounds_per_checkpoint=[100]`
- `finetune_optimzier=['sgd']`
- `finetune_max_epochs=[20]`

Tuned hyperparameters (best values are highlighted in **value**):

- `finetune_learning_rate=[**10^(-1.0)**, 10^(-0.6), 10^(-0.2), 10^(0.2), 10^(0.6), 10^(1.0)]`
- `finetune_last_layer=[True, **False**]`

### Landmarks

Fixed hyperparameters (we use the best FedAdam hyperparameters from [74]):

- `client_learning_rate=[0.01]`
- `server_learning_rate=[10^(-2.5)]`
- `server_adam_epsilon=[10^(-5)]`
- `clients_per_train_round=[64]`
- `train_batch_size=[16]`
- `total_rounds=[30000]`
- `valid_clients_per_round=[32]`
- `test_clients_per_round=[96]`
- `rounds_per_evaluation=[1000]`
- `rounds_per_checkpoint=[1000]`
- `finetune_optimzier=['sgd']`
- `finetune_max_epochs=[10]`

Tuned hyperparameters (best values are highlighted in **value**):

- `finetune_learning_rate=[0.0001, 0.001, 0.005, **0.007**, 0.01, 0.03, 0.05]`
- `finetune_last_layer=[True, **False**]`

**TedMulti-EnEs**

Fixed hyperparameters:

- `clients_per_train_round=[32]`
- `train_batch_size=[16]`
- `total_rounds=[1500]`
- `valid_clients_per_round=[98]`
- `test_clients_per_round=[117]`
- `rounds_per_evaluation=[30]`
- `rounds_per_checkpoint=[50]`
- `finetune_optimzier=['sgd']`
- `finetune_max_epochs=[20]`

Tuned hyperparameters (best values are highlighted in **value**):

- `client_learning_rate=[10^(-2.5), 10^(-2), 10^(-1.5), **10^(-1)**, 10^(-0.5)]`
- `server_learning_rate=[10^(-2.5), **10^(-2)**, 10^(-1.5), 10^(-1), 10^(-0.5)]`
- `finetune_learning_rate=[**0.0005**, 0.0007, 0.001, 0.002, 0.003]`
- `finetune_last_layer=[True, **False**]`

### F.1.2 HypCluster

Definitions of all the hyperparameters for this algorithm can be found in `hypcluster_trainer.py`. Because HypCluster with random initialization usually ends up with all clients choosing the same model (i.e., the mode collapse issue shown in Figure 5), we will use models learned by FedAvg to warmstart HypCluster. Specifically, we will run FedAvg (with the hyperparameters in Appendix F.1.1 above) for `num_warmstart_fedavg_rounds`; repeat this for `num_clusters` times, and use the models to warmstart HypCluster.

**EMNIST**

Fixed hyperparameters:

- `clients_per_train_round=[50]`
- `train_batch_size=[20]`
- `total_rounds=[1500]`
- `valid_clients_per_round=[100]`
- `test_clients_per_round=[100]`
- `rounds_per_evaluation=[100]`
- `rounds_per_checkpoint=[100]`
- `num_warmstart_fedavg_rounds=[100]`

Tuned hyperparameters (best values are highlighted in **value**):

- `client_learning_rate=[0.01, 0.05, **0.1**, 0.2]`
- `server_learning_rate=[0.0001, 0.0005, **0.001**, 0.002]`
- `server_adam_epsilon=[**0.0001**, 0.0005, 0.001, 0.002]`
- `num_clusters=[**2**, 3, 4, 5]`

**StackOverflow**

Fixed hyperparameters:

- `clients_per_train_round=[200]`
- `train_batch_size=[16]`
- `total_rounds=[1500]`
- `valid_clients_per_round=[200]`
- `test_clients_per_round=[200]`
- `rounds_per_evaluation=[100]`
- `rounds_per_checkpoint=[100]`
- `num_warmstart_fedavg_rounds=[100]`

Tuned hyperparameters (best values are highlighted in **value**):

- `client_learning_rate=[0.1, **0.5**, 1.0, 2.0]`
- `server_learning_rate=[**0.01**, 0.05, 0.1, 0.2]`
- `server_adam_epsilon=[10^(-5), **10^(-4)**, 10^(-3), 10^(-2)]`
- `num_clusters=[**2**, 3, 4, 5]`

**Landmarks**

Fixed hyperparameters:

- `clients_per_train_round=[64]`
- `train_batch_size=[16]`
- `total_rounds=[30000]`
- `valid_clients_per_round=[32]`
- `test_clients_per_round=[96]`
- `rounds_per_evaluation=[1000]`
- `rounds_per_checkpoint=[1000]`
- `num_warmstart_fedavg_rounds=[8000]`

Tuned hyperparameters (best values are highlighted in **value**):

- `client_learning_rate=[10^(-3), **10^(-2.5)**, 10^(-2), 10^(-1.5)]`
- `server_learning_rate=[10^(-3.5), **10^(-3)**, 10^(-2.5), 10^(-2)]`
- `server_adam_epsilon=[10^(-6), 10^(-5), **10^(-4)**, 10^(-3)]`
- `num_clusters=[**2**, 3, 4]`

**TedMulti-EnEs**

Fixed hyperparameters:

- `clients_per_train_round=[32]`
- `train_batch_size=[16]`
- `total_rounds=[1500]`
- `valid_clients_per_round=[98]`
- `test_clients_per_round=[117]`
- `rounds_per_evaluation=[30]`

- `rounds_per_checkpoint=[50]`
- `num_clusters=[2]`
- `num_warmstart_fedavg_rounds=[100]`

Tuned hyperparameters (best values are highlighted in **value**):

- `client_learning_rate=[10^(-2.5), 10^(-2), 10^(-1.5), **10^(-1)**, 10^(-0.5)]`
- `server_learning_rate=[10^(-2.5), **10^(-2)**, 10^(-1.5), 10^(-1), 10^(-0.5)]`
- `server_adam_epsilon=[**0.001**, 0.00001]`

### F.2 FedAvg + kNN-Per

We use the model trained by FedAvg (same as "FedAvg + Fine-tuning") as the global model for personalization. Following [54], we set the number of nearest neighbors $k$ to 10 for all experiments (the paper shows that the performance of kNN-Per is robust to this value). As mentioned in Section B.3, we tune the interpolation coefficient globally for all clients (i.e., every client use the same coefficient), because tuning per-client specific coefficient performed worse. The tuned interpolation hyperparameters are shown below (best values are highlighted in **value**):

**EMNIST**

`coefficient=[0, 0.1, 0.2, 0.3, **0.4**, 0.5, 0.6, 0.7, 0.8, 0.9, 1.0]`

**StackOverflow**

`coefficient=[0, 0.1, **0.2**, 0.3, 0.4, 0.5, 0.6, 0.7, 0.8, 0.9, 1.0]`

**Landmarks**

`coefficient=[0, 0.1, 0.2, 0.3, 0.4, **0.5**, 0.6, 0.7, 0.8, 0.9, 1.0]`

**TedMulti-EnEs**

`coefficient=[0, **0.1**, 0.2, 0.3, 0.4, 0.5, 0.6, 0.7, 0.8, 0.9, 1.0]`

#### F.2.1 Local training

Traning a local model at each client can be done by running `finetuning_trainer.py` with `total_rounds=0`. Note that what happens is that every client fine-tunes a random model (sent by the server) locally. As long as we set a large enough `finetune_max_epochs` (the best number of local epochs will be found after postprocessing the validation metrics), this will give the desired metrics where each client learns a local model without federation.

**EMNIST**

Fixed hyperparameters:

- `total_rounds=[0]`
- `valid_clients_per_round=[100]`
- `test_clients_per_round=[100]`
- `finetune_optimzier=['sgd']`
- `finetune_max_epochs=[200]`

Tuned hyperparameters (best values are highlighted in **value**):

- `finetune_last_layer=[True, **False**]`
- `finetune_learning_rate=[0.001, 0.01, **0.1**, 0.5, 1.0]`

## StackOverflow

Fixed hyperparameters:

- `total_rounds=[0]`
- `valid_clients_per_round=[200]`
- `test_clients_per_round=[200]`
- `finetune_optimzier=['sgd']`
- `finetune_max_epochs=[200]`

Tuned hyperparameters (best values are highlighted in **value**):

- `finetune_last_layer=[True, **False**]`
- `finetune_learning_rate=[0.1, **0.5**, 1.0]`

## Landmarks

Fixed hyperparameters:

- `total_rounds=[0]`
- `valid_clients_per_round=[32]`
- `test_clients_per_round=[96]`
- `finetune_optimzier=['sgd']`
- `finetune_max_epochs=[50]`

Tuned hyperparameters (best values are highlighted in **value**):

- `finetune_last_layer=[True, **False**]`
- `finetune_learning_rate=[0.0001, 0.001, **0.01**, 0.1]`

## TedMulti-EnEs

Fixed hyperparameters:

- `total_rounds=[0]`
- `valid_clients_per_round=[98]`
- `test_clients_per_round=[117]`
- `finetune_optimzier=['sgd']`
- `finetune_max_epochs=[50]`

Tuned hyperparameters (best values are highlighted in **value**):

- `finetune_last_layer=[**True**, False]`
- `finetune_learning_rate=[0.5, **1.0**, 2.0, 3.0]`

### F.3 Cross-silo

The definitions of hyperparameters can be found in `main.py` in the implementation folders for the Vehicle, School, and ADNI datasets respectively in our code, which will be open-sourced after the paper is accepted.

**Common hyperparameters.** The following hyperparameters are fixed across all cross-silo experiments:

- `client_optimizer=['sgd']`
- `server_optimizer=['fedavgm']`
- `fedavgm_momentum=[0.9]`
- `inner_epochs=[1]`
  (This is the number of local training epochs performed by a client during a round of training. For methods that train a global model, this is the number of local training epochs before returning the model update to the server.)

We focus on FedAvgM [29] for simplicity of hyperparameter tuning. Note also that the precise variable names of the hyperparameters (typefaced with `monospaced font`) may vary depending on the dataset-specific implementation.

**Computational resources.** For the Vehicle and School datasets, all experiments (including hyperparameter search) are done on 88 commodity CPU cores. For the ADNI dataset, each run is done on one GPU.

#### F.3.1 Local training

**Vehicle**

Fixed hyperparameters:

- `num_rounds=[500]`
- `clients_per_round=[23]`
- `batch_size=[64]` (client local training batch size)

Tuned hyperparameters (best values are highlighted in `**value**`):

- `client_lrs=[0.003, 0.01, **0.03**, 0.1, 0.3]`

**School**

Fixed hyperparameters:

- `num_rounds=[500]`
- `clients_per_round=[139]`
- `batch_size=[32]` (client local training batch size)

Tuned hyperparameters (best values are highlighted in `**value**`):

- `client_lrs=[0.001, 0.003, **0.01**, 0.03, 0.1]`

**ADNI**

Fixed hyperparameters:

- `num_rounds=[70]`
- `clients_per_round=[9]`

- `batch_size=[64]` (client local training batch size)

Tuned hyperparameters (best values are highlighted in **value**):

- `client_lrs=[0.001, **0.01**, 0.1]`

### F.3.2   FedAvg+Fine-tuning

**Vehicle**

Fixed hyperparameters:

- `num_rounds=[500]`
- `finetune_epochs=[100]` (max number of local epochs for fine-tuning)
- `finetune_every=[50]` (run fine-tuning every number of rounds)
- `finetune_optimizer=['sgd']`
- `clients_per_round=[23]`
- `batch_size=[64]` (client local training batch size)

Tuned hyperparameters (best values are highlighted in **value**):

- `client_lrs=[**0.003**, 0.01, 0.03, 0.1, 0.3]`
- `server_lrs=[0.1, 0.3, 1, 3, **10**]`
- `finetune_lrs=[0.003, 0.01, 0.03, 0.1, 0.3]`
  (The fine-tuning client learning rates are tuned separately for each client.)

**School**

Fixed hyperparameters:

- `num_rounds=[500]`
- `finetune_epochs=[100]` (max number of local epochs for fine-tuning)
- `finetune_every=[50]` (run fine-tuning every number of rounds)
- `finetune_optimizer=['sgd']`
- `clients_per_round=[139]`
- `batch_size=[32]` (client local training batch size)

Tuned hyperparameters (best values are highlighted in **value**):

- `client_lrs=[**0.001**, 0.003, 0.01, 0.03, 0.1]`
- `server_lrs=[0.1, 0.3, 1, **3**, 10]`
- `finetune_lrs=[0.001, 0.003, 0.01, 0.03, 0.1]`
  (The fine-tuning client learning rates are tuned separately for each client.)

**ADNI**

Fixed hyperparameters:

- `num_rounds=[70]`
- `finetune_optimzier=['sgd']`
- `clients_per_round=[9]`
- `batch_size=[64]` (client local training batch size)

Tuned hyperparameters (best values are highlighted in **value**):

- `finetune_last_layer=[True, False]`
- `server_lrs=[1, 3, **10**, 20]`
- `client_lrs=[0.0001, 0.001, **0.01**, 0.1]`
- `finetune_lrs=[0.0001, 0.001, 0.01, 0.1]`
  (When running FedAvg, the optimal client-side learning rate is 0.01. During fine-tuning, we tune client learning rates (i.e., fine-tuning learning rates) separately for each client.)

### F.3.3 HypCluster

**Vehicle**

Fixed hyperparameters:

- `num_rounds=[500]`
- `clients_per_round=[23]`
- `batch_size=[64]` (client local training batch size)

Tuned hyperparameters (best values are highlighted in **value**):

- `client_lrs=[0.003, **0.01**, 0.03, 0.1, 0.3]`
- `server_lrs=[0.1, 0.3, 1, 3, **10**]`
- `num_clusters=[2, 3, **4**]`
- `warmstart_fracs=[**0**, 0.2]`
  (The fraction of rounds for warm starting; the rest runs clustering training.)

**School**

Fixed hyperparameters:

- `num_rounds=[500]`
- `clients_per_round=[139]`
- `batch_size=[32]` (client local training batch size)

Tuned hyperparameters (best values are highlighted in **value**):

- `client_lrs=[0.001, 0.003, **0.01**, 0.03, 0.1]`
- `server_lrs=[0.1, 0.3, 1, 3, **10**]`
- `num_clusters=[2, **3**, 4]`
- `warmstart_fracs=[0, **0.2**]`
  (The fraction of rounds for warm starting; the rest runs clustering training.)

**ADNI**

Fixed hyperparameters:

- `num_rounds=[70]`
- `warm_start_rounds=[20]`
- `clustering_training_rounds=[50]`
- `clients_per_round=[9]`
- `batch_size=[64]` (client local training batch size)

Tuned hyperparameters (best values are highlighted in **value**):

- `server_lrs=[1, 3, **10**, 20]`
- `client_lrs=[0.0001, 0.001, **0.01**, 0.1]`
- `num_clusters=[**2**, 3, 4]`

### F.4 FedAvg + kNN-Per

We use the model trained by FedAvg (same as "FedAvg + Fine-tuning") as the global model for personalization. Following [54], we set the number of nearest neighbors $k$ to 10 for all experiments (the paper shows that the performance of kNN-Per is robust to this value). Similar to the cross-device setting, we also tune the interpolation coefficient globally for all clients (i.e., every client use the same coefficient). The tuned interpolation coefficients are shown below (best values are highlighted in **value**):

**Vehicle**

`coefficient=[0, 0.1, 0.3, 0.5, 0.7, **0.9**, 1.0]`

**School**

`coefficient=[0, 0.1, 0.3, **0.5**, 0.7, 0.9, 1.0]`

**ADNI**

`coefficient=[0, 0.1, 0.3, **0.5**, 0.7, 0.9, 1.0]`

#### F.4.1 MOCHA

Note that MOCHA [68] was implemented in its primal form (i.e. gradient descent update for the local personalized models, with task-relationship learning regularization) instead of the dual form since the dual was not derived for regression in the original paper. We were able to reproduce the results reported in the original paper in the primal (with an average error rate of 6.29 in Tables 5 and 7, smaller than 6.59 that was reported). In the primal, MOCHA has optimization hyperparameters such as client optimizer, client learning rates, batch sizes, etc. as with other methods.

**Vehicle**

Fixed hyperparameters:

- `num_rounds=[500]`
- `clients_per_round=[23]`
- `batch_size=[64]` (client local training batch size)

Tuned hyperparameters (best values are highlighted in **value**; an alternative combination that gives very similar results are highlighted in _values_):

- `client_lrs=[0.003, _0.01_, **0.03**, 0.1, 0.3]`
- `lambdas=[**0.0001**, _0.001_, 0.01, 0.1, 0.3, 1, 3]`
  (MTL regularization strength.)
- `mocha_outers=[**1**, _2_, 5]`
  (The number of local epochs every server update of the task-relationship matrix. This nests with `inner_epochs` and may thus increase the total number of epochs over the local datasets.)

**School**

Fixed hyperparameters:

- `num_rounds=[500]`
- `clients_per_round=[139]`
- `batch_size=[32]` (client local training batch size)

Tuned hyperparameters (best values are highlighted in `**value**`; an alternative combination that gives very similar results are highlighted in `_values_`):

- `client_lrs=[0.001, **0.003**, _0.01_, 0.03, 0.1]`
- `lambdas=[**0.0001**, 0.001, _0.01_, 0.1, 0.3, 1, 3]`
  (MTL regularization strength.)
- `mocha_outers=[_1_, **2**, 5]`
  (See Vehicle hyperparameters above for description.)

### F.4.2 Ditto

**Vehicle**

Fixed hyperparameters:

- `num_rounds=[500]`
- `personalized_model_inner_epochs=[1]`
- `clients_per_round=[23]`
- `batch_size=[64]` (client local training batch size)

Tuned hyperparameters (best values are highlighted in `**value**`):

- `client_lrs=[0.003, 0.01, 0.03, 0.1, **0.3**]`
- `server_lrs=[0.1, 0.3, **1**, 3, 10]`
- `lambdas=[0.0001, 0.001, **0.01**, 0.1, 0.3, 1, 3]`
  (MTL regularization strength.)
- `personalized_model_lrs=[0.003, 0.01, 0.03, **0.1**, 0.3]`
  (The client learning rate for Ditto's personalized models. This may be different from the client learning rates used to update the global model.)

**School**

Fixed hyperparameters:

- `num_rounds=[500]`
- `personalized_model_inner_epochs=[1]`
- `clients_per_round=[139]`
- `batch_size=[32]` (client local training batch size)

Tuned hyperparameters (best values are highlighted in `**value**`):

- `client_lrs=[**0.001**, 0.003, 0.01, 0.03, 0.1]`
- `server_lrs=[0.1, 0.3, 1, 3, **10**]`
- `lambdas=[0.0001, 0.001, 0.01, 0.1, 0.3, **1**, 3]`
  (MTL regularization strength.)
- `personalized_model_lrs=[0.001, 0.003, **0.01**, 0.03, 0.1]`
  (See Vehicle hyperparameters above for description.)

**ADNI**

Fixed hyperparameters:

- `num_rounds=[100]`
- `clients_per_round=[9]`
- `local_batch_size=[64]`
- `personalized_model_local_epoch=[1]`

Tuned hyperparameters (best values are highlighted in `**value**`):

- `server_lrs=[1, 3, **10**, 20]`
- `client_lrs=[0.0001, 0.001, **0.01**, 0.1]`
- `personalized_model_lrs=[0.0001, **0.001**, 0.01]`
  (See Vehicle hyperparameters above for description.)
- `lambdas=[0.01, 0.1, **1**]`
  (MTL regularization strength.)

## G   Additional Experimental Results and Discussions

In this section, we provide a few more experimental results that are omitted from the main paper due to space limitation.

**Cross-device experiments.** Table 6 extends Table 4 with standard deviations across 5 different runs for each metric. Note that there are two standard deviations here: 1) the standard deviation of per-client accuracy across the test clients, which is used as a fairness metric[7] and is already included in Table 4; 2) the standard deviation of each metric value (e.g., the standard deviation of the fairness metric mentioned in 1)) across 5 different runs (each run uses a different seed), which is omitted from the main paper and is included in Table 6.

**Cross-silo experiments.** Similar to the cross-device counterpart, Table 7 extends Table 5 with standard deviation across 5 different runs for each metric. One small caveat around the use of client-specific fine-tuning hyperparameters, as discussed in Appendix C of the main paper, is that it may improve performance only when the clients' local datasets are large. Here, we perform an additional analysis on the School dataset, where the local datasets are relatively small (recall Table 2). Figure 11 visualizes the statistics of clients hurt when training on School, akin to Figure 8 (note that we use MSE here, so a positive delta means this client gets hurt by fine-tuning). Observe that while many clients benefited from the client-specific fine-tuning (those with negative metric deltas), some clients do not observe an improvement and may even see a slight degradation in their utility compared to the client-agnostic fine-tuning setting. We argue that this is due to clients having small local datasets such that the selected client-specific FT hyperparameters may overfit to the small local validation sets and may not reflect general improvement on the local test sets. Note also that while Table 5 and Table 7 suggests a seemingly considerable fraction (33%) of clients is hurt after fine-tuning, we can observe from Figure 11 that most 'hurt' clients have borderline metric changes. Overall, these observations suggest that the effect of client-specific fine-tuning can be mixed if the client local datasets are relatively small, as is usually the case with cross-device federated learning.

**Table 6:** Complete results of the experiments in Table 4. The only difference is that we report each metric's mean and standard deviation (std) across 5 different runs. Note that there are two stds here: 1) "per-client acc std", the std of per-client accuracy across the test clients, which is a fairness metric considered in [49, 53] and is already included in Table 4; 2) the std across 5 different runs, which is omitted from Table 4 and is shown after ± in this table.

| Algorithm | Metrics | EMNIST | StackOverflow | Landmarks | TedMulti |
|---|---|---|---|---|---|
| Local | Per-client acc mean | 0.594±.011 | 0.062±.001 | 0.173±.008 | 0.056±.0004 |
| | Per-client acc std | 0.17±.011 | 0.02±.005 | 0.16±.008 | 0.02±.0003 |
| FedAvg + Fine-tuning (FT) | Per-client acc mean before FT | 0.844±.004 | 0.269±.002 | 0.564±.008 | 0.160±.002 |
| | Per-client acc std before FT | 0.1±.004 | 0.03±.0007 | 0.16±.005 | 0.04±.004 |
| | Per-client acc after FT | 0.903±.002 | 0.282±.002 | 0.773±.006 | 0.162±.002 |
| | Per-client acc std after FT | 0.06±.0005 | 0.03±.001 | 0.11±.006 | 0.04±.001 |
| | % clients "hurt" after FT | 5.2%±0.7% | 14%±2.4% | 5.6%±2.4% | 40%±4.8% |
| | FT all layers vs last layer | All layers | All layers | All layers | All layers |
| | *Practical concerns*: difficult to tune hyperparameters; may hurt clients; sensitive to distribution shift; performance drops with fewer local examples (see Section B.1) | | | | |
| HypCluster / IFCA | Per-client acc mean | 0.897±.002 | 0.273±.0009 | 0.573±.005 | 0.163±.004 |
| | Per-client acc std | 0.08±.003 | 0.03±.002 | 0.16±.003 | 0.04±.002 |
| | No. tuned clusters ($k$) | 2 | 2 | 2 | 2 |
| | % clients largest cluster | 52.6%±0.8% | 85.1%±1.6% | 92.1%±3.6% | 54.7%±0 |
| | Warmstart from FedAvg | Yes | Yes | Yes | Yes |
| | Per-client acc mean by ensembling $k$ FedAvg models | 0.860±.005 | 0.271±.002 | 0.564±.016 | 0.163±.004 |
| | Per-client acc std by ensembling $k$ FedAvg models | 0.08±.006 | 0.03±.002 | 0.16±.007 | 0.04±.004 |
| | *Practical concerns*: difficult to train due to mode collapse; high communication cost; difficult to interpret the learned clusters; sensitive to distribution shift (see Section B.2) | | | | |
| FedAvg + kNN-Per | Per-client acc mean | 0.876±.003 | 0.275±.0005 | 0.735±.007 | 0.162±.002 |
| | Per-client acc std | 0.06±.004 | 0.03±.0009 | 0.13±.005 | 0.05±.004 |
| | % clients "hurt" | 19.8%±4.2% | 23.6%± 3.4% | 5.6%±2.3% | 34.4%±4.5% |
| | *Practical concerns*: difficult to tune hyperparameters; may hurt clients; sensitive to distribution shift; performance drops with fewer local examples (see Section B.3) | | | | |

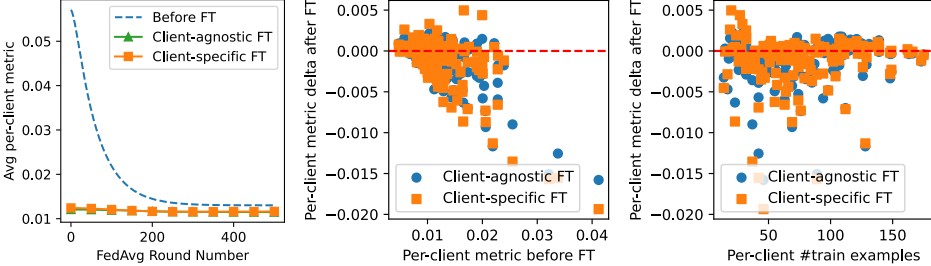

**Figure 11:** Effects of client-specific fine-tuning on the School dataset (each client chooses a custom fine-tuning learning rate and stopping epoch). Note that School uses MSE (mean squared error) as the evaluation metric here, thus the **more negative** the per-client metric delta, the better. Unlike Figure 7, tuning the fine-tuning hyperparameters globally and tuning in a per-client manner performs the same on the School dataset. This is potentially due to the fact that the client's local dataset is relatively small for School (as shown in Table 2).

**Table 7:** Complete results of the experiments in Table 5. The only difference is that we report each metric's mean and standard deviation (std) across 5 different runs. Similar to results in cross-device experiments, there are two stds here: 1) "per-client acc std", the std of per-client accuracy across the test clients, which is a fairness metric considered in [49, 53] and is already included in Table 5; 2) the std across 5 different runs, which is omitted from Table 5 and is shown after $\pm$ in this table.

| Algorithm | Metrics | Vehicle (acc) | School (mse) | ADNI (mse) |
|---|---|---|---|---|
| Local training | Per-client metric mean | 0.9367±.0248 | 0.0121±.0059 | 0.0177±.0007 |
| | Per-client metric std | 0.0027±.0029 | 0.0003±.0008 | 0.0106±.0007 |
| FedAvg + Fine-tuning (FT) | Per-client metric mean before FT | 0.8859±.0833 | 0.0130±.0068 | 0.0141±.0004 |
| | Per-client metric std before FT | 0.0033±.0028 | 0.0002±.0009 | 0.0091±.0012 |
| | Per-client metric mean after FT | 0.9385±.0253 | 0.0116±.0056 | 0.0124±.0007 |
| | Per-client metric std after FT | 0.0029±.0015 | 0.0002±.0006 | 0.0082±.0012 |
| | % clients "hurt" after FT | 4.4%±3.9 | 33.0%±3.0 | 0±0 |
| | FT all layers vs last layer | N/A | N/A | no difference |
| | *Practical concerns: Similar to those in the cross-device setting (Table 4); Tuning per-client FT hyperparameter may outperform tuning globally (Figure 7).* | | | |
| HypCluster / IFCA | Per-client metric mean | 0.9246±.0288 | 0.0112±.0053 | 0.0137±.0012 |
| | Per-client metric std | 0.0058±.0043 | 0.0003±.0006 | 0.0093±.0017 |
| | No. tuned clusters ($k$) | 4 | 3 | 2 |
| | % clients largest cluster | 49.6%±3.5 | 44.6%±1.9 | 78%±17.6 |
| | Warmstart from FedAvg | No | Yes | Yes |
| | Per-client metric mean by ensembling $k$ FedAvg models | 0.8851±.0828 | 0.0129±.0066 | 0.0133±.0006 |
| | Per-client metric std by ensembling $k$ FedAvg models | 0.0028±.0032 | 0.0001±.0008 | 0.0091±.0011 |
| | *Practical concerns: Similar to those in the cross-device setting (Table 4); on Vehicle and School, mode collapse may occur less potentially because of linear model.* | | | |
| FedAvg + kNN-Per | Per-client metric mean | 0.9228±.0028 | 0.01163±.0002 | 0.0126±.0008 |
| | Per-client metric std | 0.0287±.0016 | 0.0055±.0006 | 0.0096±.0006 |
| | % clients "hurt" | 22.6%±4.3% | 37.4%±0.6% | 37.8%±16.8% |
| | *Practical concerns: Similar to those in the cross-device setting (Table 4).* | | | |
| MTL (Ditto) | Per-client metric mean | 0.9377±.0218 | 0.0114±.0053 | 0.0134±.0004 |
| | Per-client metric std | 0.0025±.0026 | 0.0002±.0006 | 0.0074±.0008 |
| MTL (Mocha) | Per-client metric mean | 0.9371±.0244 | 0.0121±.0059 | N/A |
| | Per-client metric std | 0.0030±.0025 | 0.0003±.0009 | N/A |

# H    Conclusion

We present `Motley`, the first large-scale benchmark of personalized federated learning covering both cross-device and cross-silo settings. `Motley` provides an end-to-end experimental pipeline including data preprocessing, algorithms, evaluation metrics, and tuned hyperparameters, which ensures reproducibility. Beyond these baselines, our experiments provide new insights about personalized FL and suggest several directions of future work.

- The notion of the "best" method (or even the "best" hyperparameter of the same method) can change when we change the evaluation metrics or settings. In the cross-device setting, FedAvg+Fine-tuning achieves the best average per-client accuracy for 3/4 datasets, and at the same time also improves fairness[7] (Table 4 and Figure 2). The best hyperparameters of FedAvg+Fine-tuning can change if we look at different metrics (Figure 9). On the TedMulti-EnEs dataset, it is difficult to determine which personalization algorithm is the "best": while HypCluster achieves the best average per-client accuracy, it is worse than FedAvg+Fine-tuning from the perspective of accuracy-communication ratio (Figure 5(c)); on the other hand, the fine-tuned models of $40\%$ clients on TedMulti-EnEs are worse than that of FedAvg (Table 4). In the cross-silo setting, both FedAvg+Fine-tuning and MTL achieve the best average per-client accuracy over the three datasets (Table 5). MTL may have an additional advantage of having less hyperparameters than FedAvg+Fine-tuning. On the Vehicle dataset, local training seems the "best" because it achieves a similar accuracy as other methods but is more private. Given that the notion of "best" method can change depending on the metrics or settings, a critical future direction is thus to develop systematic evaluation schemes for personalized FL (i.e., mean accuracy alone is not enough).

- Existing literature often overlook or obfuscate the practical complexities of deploying personalized FL algorithms in real-world settings. For example, in Section B.1, we discuss that local data scarcity and heterogeneity create a fundamental challenge to tuning FedAvg+Fine-tuning hyperparameters in the cross-device setting. Because each client has a very small local dataset in the cross-device setting, the fine-tuning hyperparameters are tuned globally (instead of in a per-client manner as in the cross-silo setting). The globally tuned hyperparameters may hurt some clients (Figure 8). Ideally, we want to find a good set of fine-tuning hyperparameters such that the overall improvement is large and no clients are hurt after fine-tuning, which can be fundamentally difficult (Figure 9).

- Improving HypCluster requires solving the mode collapse issue and rethinking the difference between clustering and ensembling. How to effectively train HypCluster in the real-world federated learning systems is an interesting open problem (Figure 5(a)). Although warmstart can mitigate the mode collapse issue, Table 4 and Table 5 show that the performance of HypCluster is similar to that of ensembling[20] multiple models learned by FedAvg. Does HypCluster really capture the underlying clustering structure (Figure 5(d))? Answering this question requires interpreting the learned clusters in the federated learning setting.

- Tradeoffs exists between adapting a client's personalized model to the current local distribution and generalizing to future distributions. As shown in Figure 4, compared to FedAvg, the fine-tuned models are more sensitive to the distribution shift between the examples used in fine-tuning and testing. As a result, it may be necessary to continuously fine-tune the model when clients have new data. Besides the fine-tuning method, this tradeoff between personalization and generalization exists for other personalized FL algorithms as well, which is worth exploring in greater detail.

- Given the observed benefits of per-client hyperparameter tuning in cross-silo FL (Figure 7), it may be beneficial to develop similar, scalable approaches for hyperparameter tuning in cross-device FL.

**Limitations and future work.** `Motley` can be expanded in the following ways. First, add more evaluation metrics. Our benchmark results contains the following metrics (see Table 4 and Table 5): the average per-client accuracy, fairness[7] (clients should have similar local accuracies), and algorithm-specific metrics such as the percentage of clients hurt by fine-tuning. When comparing algorithms on specific datasets, we also use metrics such as the communication cost (Figure 5) and robustness to distribution shift (Figure 4). `Motley` can be expanded by adding more evaluation metrics such as privacy, other notions of fairness[7], and robustness to attacks. Second, add more datasets, especially those with known clustering structure (similar to TedMulti-EnEs) and datasets representing real-world cross-silo applications. Third, add more algorithms. `Motley` currently has five model-agnostic federated learning personalization algorithms. Expanding `Motley` to include more algorithms (see Appendix D) would be another important future work.

