# OpenReview forum: "Motley: Benchmarking Heterogeneity and Personalization in Federated Learning"
_NeurIPS.cc/2022/Workshop/Federated_Learning — FL-NeurIPS 2022 Poster_

### Official Review · Reviewer_gWbX · 2022-10-17
**Review of Motley**

### Summary

This paper introduces Motley, a benchmarking suite for personalised Federated Learning in the cross-device and cross-silo settings. It integrates five personalisation techniques (fine-tuning, HypCluster, knn-personalisation, Multi-task learning) over seven datasets and showcases results, along with interesting take-away messages about the applicability and success of personalised FL.

### Strengths

* The paper is generally well-written and easy to follow and extract information. The experimental setup is clearly laid out in the appendix and reproducibility will be further enhanced when the authors open-source their codebase.
* The authors integrate various settings, datasets, models and metrics to evaluate and extract insights in a multi-dimensional space (accuracy, fairness, clustering, communication cost).
* I particularly liked the extracted insights about what works and what doesn't in the field of federated personalisation.

### Weaknesses

* The paper does not introduce novel ideas, nor does it strongly differentiates from prior work, apart from the breadth of analysis, such as including both cross-silo and cross-device settings.
* The paper completely disregards the system heterogeneity in the cross-device setting, an integral characteristic of such deployments.
* The paper misses references to alternative aggregators and architectural techniques that may be tackling personalisation in different ways.

### Further comments

* The experimental setup is missing the frameworks and types of hardware used.
* While the authors are making observations about the behaviour of several personalisation algorithms and their pathogenies, they do not propose any methods that solves/mitigates them in the paper.
* The authors mention in the introduction that Motley is a useful tool for the areas of transfer, meta and multi-task learning, but I do not see the benefit from FL upstream. Maybe the authors could clarify how this work helps in that dimension.
* Clients seems to be making different amounts of work, based on a fixed local epoch configuration, but varying dataset sizes. This could further exacerbate the existence of stragglers.
* I am wondering what is the fairness behaviour of clients participating in global training and further personalise vs. those who do not participate in the former phase but still personalise.
* Maybe instead of finetuning the whole model and keeping two variants (Remark of 132), the same model could be partially finetuned by means of training some parameters only (e.g. [a]).
* Missing personalisation/mtl techniques, such as pFedMe [b], FedEm [c], FedPNAS [d]

[a] Leontiadis, Ilias, et al. "It's always personal: Using early exits for efficient on-device CNN personalisation." Proceedings of the 22nd International Workshop on Mobile Computing Systems and Applications. 2021.
[b] Canh T. Dinh, Nguyen H. Tran, and Tuan Dung Nguyen. Personalized federated learning with moreau envelopes. In Advances in Neural Information Processing Systems 33: Annual Conference on Neural Information Processing Systems, 2020.
[c] Othmane Marfoq, Giovanni Neglia, Aurélien Bellet, Laetitia Kameni, and Richard Vidal.
Federated multi-task learning under a mixture of distributions.
[d] Hoang, Minh, and Carl Kingsford. "Personalized Neural Architecture Search for Federated Learning." (2021).

### Questions

* The authors state that the globally tuned hyperparameters can hurt finetuning. Wouldn't the same case apply in the global training phase?
* Would personalisation benefit from centrally-pretrained global models, similar to [e]?
* Would it be possible for different personalisation techniques to be used in conjunction with one another?

[e] Qu, Liangqiong, et al. "Rethinking architecture design for tackling data heterogeneity in federated learning." Proceedings of the IEEE/CVF Conference on Computer Vision and Pattern Recognition. 2022.

### Presentation

* Tables would need to be reworked and made consistent in terms of formatting and easier to read.
    * In the same direction, the best results could be rendered in bold where applicable.
* Figure 1: ADNI would benefit from having markers. In reality, it is much more discrete that shown to be (9 clients).
* Figure 3c: The authors could consider using a different ticks to illustrate the actual learning rates used.
* L115, L138: typos
* I am not sure if checklist is needed
* Given the appendix is quite lengthy, maybe it would benefit from a table of contents or an introduction which lays out what is where.

---

### Official Review · Reviewer_iPPR · 2022-10-17
**The paper proposes a benchmark for personalized federated learning called Motley. Motley evaluates a variety of datasets across a suite of cross-device and cross-silo federated learning systems using evaluation metrics for better understanding the possible impacts of personalization as well as accuracy and fairness. They make some interesting observations that warrant further research and thus open the grounds for more work in this field.**

Strengths -
1.	They try to better understand the problems of FL personalization, which is an under-studied yet important field of research.
2.	They have a good variety of datasets, systems and evaluation metrics.
3.	They make some interesting observations and provide decent commentary on them which does indeed promote further research into these areas.

Weaknesses -
1.	Explanations of the metrics and how they are used for the different aspects of personalization are missing.
2.	Some results are not fully explained and seem to counter each other
3.	Data heterogeneity is not used sufficiently

Detaions -
1.	The addition of the stateless/stateful properties to distinguish cross-device and cross-silo is confusing and unnecessary. It does not add anything to the observations.
2.	As with fairness, the definition of “state” should also be specified by the authors. For example, in cross-device, the global model is in fact “stateful” since it is updated between rounds. It is the clients that are not required to be “stateful”, but it is also acceptable if they are.
3.	For fine-tuning, it is understandable that the models overfit on local data. However, would that not decrease the fairness metric values? Why do we instead see lower variance in Figure 2?
4.	What is the impact of OOD on fairness?
5.	For FL analysis, there must be evaluations with varying data heterogeneity since it is one of the essential and most impactful properties of FL. No results were presented in this aspect and leaves many questions unanswered. This is a critical addition required for this paper.

---

### Official Review · Reviewer_qB5x · 2022-10-18
**Motley:  Benchmarking Heterogeneity and Personalization in Federated Learning**

This paper proposes Motley, a benchmark for personalized federated learning. The main contribution of this paper is adding personalization module (including training and evaluation) to the existing fl benchmark. The authors evaluate different personalization techniques in four cross-device and three cross-silo federated datasets. The experimental results highlight strengths and weaknesses of existing personalization approaches.

Strengths:
+ Take cross-device and cross-silo fl setting into account.
+ Add personalization module (including training and evaluation) to the fl benchmark.
+ Paper is well organized and well written.
+ While I recognize the low novelty of the work, I believe that it is a solid step towards the personalization in federated learning.

Weaknesses:
- The technical contribution of this paper is limited. This paper did not propose new methods or other novel contributions. A more significant contribution would be propose a better personalization evaluation metrics.
- The results dont bring many insights from the experiments. Many of the observations are already well known.
- There is no clear difference between the strengths and weakness of the chosen personalization approaches. What are the best fitting suitations for each approach?

---

### Decision · Program_Chairs · 2022-10-20

Accept (Poster)